# 12.1 terabit/second data center interconnects using O-band coherent transmission with QD-MLL frequency combs

Santiago Bernal [1] ✉, Mario Dumont[2], Essam Berikaa [1], Charles St-Arnault[1], Yixiang Hu [1], Ramon Gutierrez Castrejon [1,3], Weijia Li [1], Zixian Wei[1], Benjamin Krueger[4], Fabio Pittalà[4], John Bowers [2] & David V. Plant[1]

Most current Data Center Interconnects (DCI) use intensity modulation direct detection (IMDD) configurations due to their low complexity and cost. However, significant scaling challenges allow coherent solutions to become contenders in these short reach applications. We present an O-band coherent optical fiber transmission system based on Quantum Dot−Mode Locked Lasers (QD-MLLs) using two independent free-running comb lasers, one each for the carrier and the Local Oscillator (LO). Using a comb-to-comb configuration, we demonstrate a 10 km single mode fiber, O-band, coherent, heterodyne, 12.1 Tbps system operating at 0.47 Tbps/λ using 26 λs. We used fewer comb lines (26 λs), faster symbol rate (56 GBaud) and higher constellation cardinality (32 QAM) relative to the highest capacity C-band systems reported to date. Through design, analysis, and experimentation, we quantify the optimum comb line spacing for this use case. We compare potential configurations for increasing data center interconnect capacities whilst reducing power consumption, complexity, and cost.

Thousands of Terabits per second propagate on the optical fiber transmission networks used to deliver Internet connectivity. Data Centers (DCs) constitute an integral part of this interconnect infrastructure. Current and forecasted Data Center Interconnect (DCI) performance requirements project that Tbps capacities, deployed over <10 km Single Mode Fiber (SMF), are necessary for both intra- (~0.5–2 km), and inter- (~2–10 km SMF) DCI segments[1,2]. This is because 70% of all Internet traffic occurs inside a DC[3]. Problematically, DCs are responsible for 1–2% of global electricity consumption, with projections to reach 20% by 2025[4].

Currently deployed DCI optical transceivers are built using Intensity Modulation/Direct Detection (IMDD), and Pulse Amplitude Modulation (PAM) modulation formats (e.g., 800 Gbps using PAM4)[5,6] transmitted in the O-band. These systems provide low cost, power consumption, latency, and complexity solutions. However, increasing

IMDD transceiver capacities is problematic and constrained (details below in "IMDD")[7]. Alternatively, coherent transceivers are capable of (i) scaling to higher capacities, (ii) meeting DCI reach requirements, and (iii) delivering a lower power, cost-effective alternative to IMDD[8–12]. Both architectures also support the use of Wavelength Division Multiplexing (WDM) to increase capacity. These have been mostly demonstrated using arrays of ECLs. They can be implemented using SMF or multicore fibers and can achieve high rates. Additionally, very broadband optical spectrums can be used over long distances at the cost of higher complexity and number of components. Recent demonstrations of high throughput systems over longer distances have been achieved using multicore fibers[13–15] and ultrawide and/or multi-band bandwidths[16–19]. The array of ECLs recently started being replaced by more efficient multi-λ laser sources such as combs for both IMDD[20] and coherent[21] demonstrations.

[1]Department of Electrical and Computer Engineering, McGill University, Montreal, QC H3A 0G4, Canada. [2]Department of Electrical and Computer Engineering, University of California, Santa Barbara, CA 93106, USA. [3]Inst. of Engineering, Univ. Nacional Autónoma de México UNAM, Cd. Uni., 04510 Mexico City, Mexico. [4]Keysight Technologies Deutschland GmbH, Böblingen 71034, Germany. ✉e-mail: Santiago.Bernal@mail.mcgill.ca

In this paper, we present the an O-band coherent optical fiber transmission system based on Quantum Dot−Mode Locked Lasers (QD-MLLs) using two independent free-running comb lasers, one each for the carrier and the Local Oscillator (LO). Using a comb-to-comb configuration, we demonstrate a 10 km SMF, O-band, coherent, homodyne, 12.1 Tbps system operating at 0.47 Tbps/λ using 26 λs. As shown in Fig. 1, we used 30 fewer comb lines (26 vs. 56 λs), twice the baud rate (56 vs. 28 GBaud) and double the constellation cardinality (32 QAM vs. 16 QAM) relative to the highest capacity comb based C-band systems reported to date[18,22–28]. This was possible due to the higher spacing between lines allowing for higher symbol rates and the lower phase noise and stability of the laser allowing for higher modulation formats. Our work examines a QDot-MLL demonstration using both a comb as carrier and LO while also providing a ~3-fold improvement on previous QDash-MLL comb-to-comb coherent demonstrations[27,29]. Our results suggest O-band comb-based links can provide a scalable solution for simpler DCI systems. For Fig. 1, we restricted the comparison to comb laser demonstrations that that employ (i) a single comb laser for the carrier, and (ii) propagation over SSMF.

## IMDD

Present day 400/800 Gbps DCI optical fiber transmission systems (OFTS) use: (i) O-band, IMDD, and Coarse Wavelength Division Multiplexing (CWDM) with either 4 or 8 channels spaced by 20/10 nm and operating at 100/200 Gbps/λ[6]; or (ii) C-band, coherent, and single channel operating at 800 Gbps/λ[30]. The next capacity node on the Ethernet 802.3 Roadmap for deployment in 2024 is 1.6 Tbps[2,31]. In parallel commercial O-band, IMDD, 1.6 Tbps deployments (2023/2024) are implemented using an 8 x 200G architecture or a 16 x 100G architecture[32,33]. Commercially, 5 and 3 nm CMOS nodes are viable options to achieve 1.6 Tbps operation. For example, MARVELL's Nova™ 1.6 T IMDD PAM4 DSP engine is implemented using 5 nm CMOS[32], and Ciena's WaveLogic 6 coherent transceivers utilize 3 nm CMOS[2].

IMDD scalability is limited by chromatic dispersion, forcing higher throughputs to be achieved by increasing spatial parallelism and/or the number of wavelengths. Unfortunately, Parallel Single Mode (PSM) IMDD solutions can not scale because fiber counts become untenable. For example, 1.6 bps can be achieved by using two fibers each carrying 4 × 200 Gbps or 8 × 100 Gbps or a single fiber with 4 × 400 Gbps[34]. However, the increase to 12.8 Tbps would require 16 fibers operating at 4 × 200 Gbps each. 16 fibers, or more, start to become untenable for practical reasons (e.g., pluggable faceplate densities, fiber costs).

Alternatively, capacity scaling is achievable using the CWDM $\Delta\lambda = 20$ nm grid (center-to-center $\lambda = 20$ nm) and increases in baud rate and/or modulation format. At the edge wavelengths of 1270 nm and 1310 nm, and beyond ~200 Gbps/λ, implementation with 112 Gbaud/PAM4 signaling, is limited by chromatic dispersion to ≤2 km reach[35,36]. Therefore, a move to higher baud rates is prohibitive because of chromatic dispersion. Also, impractical SNR requirements limits scaling[37,38]. Therefore, the benefits of increasing IMDD signaling are marginalized because powerful DSP engines are needed.

IMDD capacity increases can be achieved by decreasing the wavelength spacing while managing the chromatic dispersion limitations. The decreased wavelength spacing scenario generates stringent specification requirements for the optical Mux/DeMux pair required at

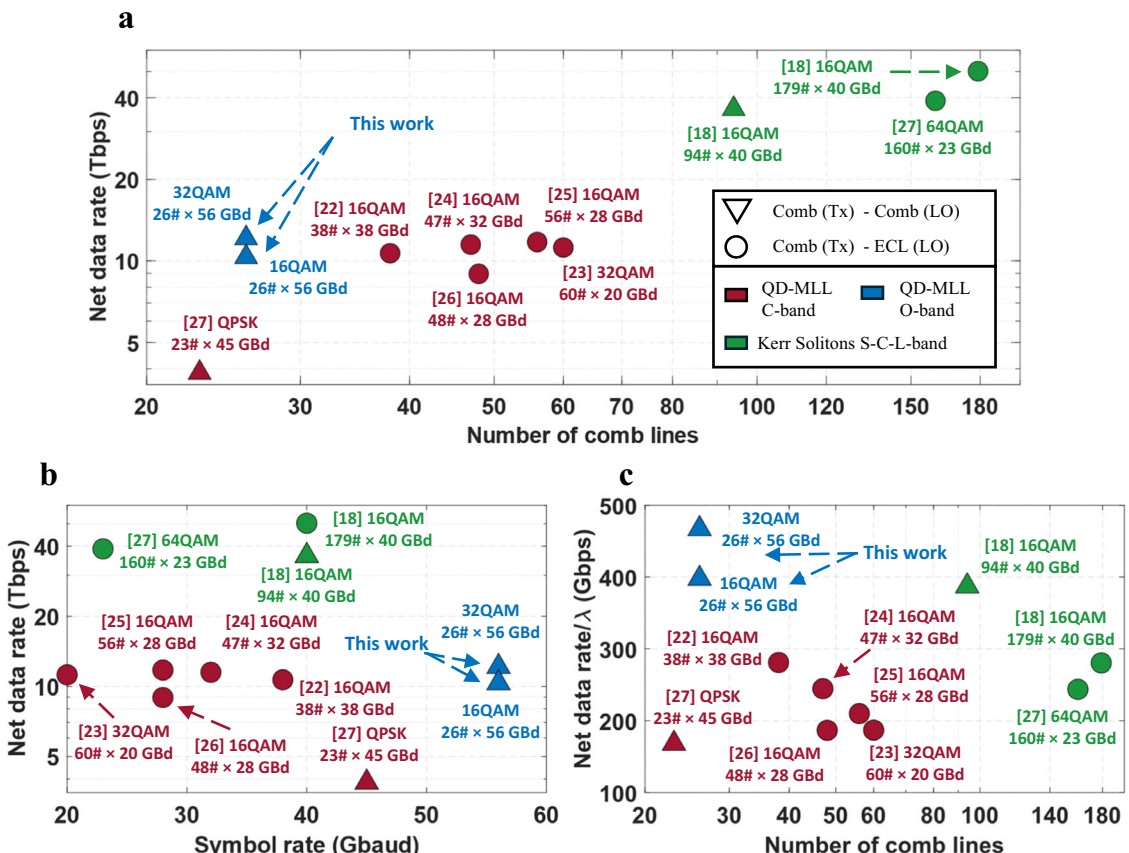

**Fig. 1 | Review of recent coherent transmission system experiments.** Demonstrations using a Quantum Dash/Dot Mode Locked Laser (QD-MLL) comb as carrier and either another comb or an External Cavity Laser (ECL) as Local Oscillator (LO). **a** Net data rate and number of comb lines. **b** Net data rate and symbol rate. **c** Net data rate per λ and number of comb lines. The legend applies to all three plots.

the transmitter and receiver, respectively (e.g., narrower filter responses, sharper filter edge performance) because the Mux/DeMux Δλ response must be tightly controlled (see Supplementary Information S5). Additionally, IMDD can suffer large nonlinearity penalties such as four-wave mixing (FWM) when multiple channels are used as seen in "Nonlinearities in comb laser based OFTS". Active laser tuning and stabilization using Themo Electric Coolers (TECs) is a required consequence of this approach.

Alternatively capacity scaling via per λ baud rate and QAM format increases are achievable using coherent transceivers[39]. Challenges to be addressed before deploying coherent transceivers in the DCI segment are centered on developing and demonstrating low power, low complexity, and cost effective solutions[40,41]. Here we present a reference solution for the 10 km or less DCI segment utilizing coherent transduction and Quantum Dot−Mode Locked Lasers (QD-MLLs) operating as independent and free-running carrier and LO sources.

## O-band coherent

IMDD receivers do not require the DSPs responsible for carrier recovery, including Frequency Offset (FO) and Carrier Phase (CP) estimation and correction[6]. On the other hand, coherent C-band DSP stacks include (i) chromatic dispersion (CD) compensation, and (ii) carrier recovery for each operating wavelength. The O-band comb-to-comb coherent DSP stack does not require a CD block or FO estimation (more below on this topic) for each channel. Adaptive equalization can tolerate the chromatic dispersion across the O-band over 10 km SMF without dedicating a DSP block (static filter) for this process, as the absolute dispersion is limited to <30 ps/nm at 100 Gbaud[42]. In our work, we can operate across the O-band range of 1280−1320 nm without CD compensation at 56 Gbaud. Operating in the O-band also relaxes the linewidth requirements and enables employing lasers with linewidths over 1 MHz with negligible performance penalty[31].

Additionally, a single FO estimation can be done to determine the FO between all channel pairs. This additional DSP simplification, that is intrinsic to the comb-based approach, could be applied after initialization by using a coarse frequency offset estimation on a single comb line; whereas the same FO will be constant for all comb lines if both carrier and LO have a known comb spacing. During the initial setup of the link, the FO of the central channel-pair could be measured and used to find the FO of each individual channel-pair. By continuously measuring only one channel, the correct FO correction can be applied to all channels. Matching the spacing of the carrier and LO comb sources is feasible to below 1 GHz (see Supplementary Information S4). Hence, the coarse FO estimation and the active tuning of the laser source can be performed on only one comb line, while fixing the entire spectrum. In addition, the adaptive equalizer responsible for the polarization tracking can remove the residual FO up to 20 MHz (See "QD-MLL as local oscillator"), relaxing the FO margin of error across all channel-pairs.

The O-band coherent configuration can produce a 10% reduction in the DSP power consumption when compared to a C-band coherent transceiver[43] because of the dispensed CD compensation block. The ASIC engine typically consumes 50% of the coherent transceiver module power, out of which 30% is consumed by the coherent DSP, excluding the digital-to-analog (DAC) and analog-to-digital (ADC)[44]. Thus, by removing the CD compensation block, we can potentially reduce the overall coherent transceiver module power by 3%. The power consumption of the CD compensation module in a typical C-band coherent pluggable uses frequency domain equalizers to compensate for the chromatic dispersion without using an extensively large time-domain filter. For a 56 Gbaud and 10 km transmission, equalizing the CD in the time domain requires 21 taps[45]. Although implementing 21 taps is feasible, it still consumes considerable power due to the complexity of the linear convolution. The frequency domain CD compensation block consumes 20% of the DSP power

consumption and takes up >15% of the Die area[43]. By removing this area, the DSP architecture can be reoptimized to make use of the liberated space [ref]. An additional reduction is feasible because QD-MLL comb sources require only one laser driver and TEC controller for stabilization. Hence, the estimated power saving in ~2−3.25 Watt for a 400 G/800 G transceivers[44,46].

## Comb-based DCIs

Two comb lasers can support an n-λ (for optimum n value see 5. Discussion), bidirectional, 10 km SMF fiber pair link using only two laser drivers, and two TECs, one per comb as seen in Fig. 2a. By contrast, a conventional n- λ coherent system requires one discrete laser per λ, acting as carrier and LO per λ, respectively. Therefore, n-laser drivers and n-TECs are needed which greatly increases the total power consumption of the system.

Figure 2b illustrates the concept of an integrated chip enabling comb-to-comb OFTS using QD-MLLs, dual polarization IQ modulators (DP-IQM), coherent receivers, and semiconductor optical amplifiers (SOA) fully integrated to form a Photonic Integrated Circuit (PIC). Conceptually, the figure represents a Coherent-Transmit Receiver Optical Sub Assembly (C-TROSA) and would only require a single TEC to stabilize the comb laser source used as both carrier (outbound) and LO (inbound). Here the one comb output is de-multiplexed and modulated by n-DP-IQMs being driven by n-DACs and drivers. The signals are next multiplexed and amplified by a SOA to compensate for the optical power attributed to each comb λ, and the DP-IQM modulation loss. The second QD-MLL output is used as the LO for the inbound signals. These inbound signals are amplified by the SOA, de-multiplexed and sent to n-coherent receivers for transduction before being delivered to n-ADCs.

This concept is limited by the potential high replacement cost of a single failure in such a densely integrated PIC. This risk can be mitigated by using more mature and reliable standard components as further detailed in "Discussion". This would increase the production yield and shelf-life of such a scheme. On the other hand, this risk can be removed by operating with more commercially available components such as optical pluggable transceivers and receivers which already dominate the DCI space. This concept is described in Supplementary Information S3. This would allow for individual pluggables to be replaced as needed in case of failures.

## Results

Optical frequency combs are a viable approach to circumvent the need to install more parallel lasers acting as a modulated source (EML/DML), or an indirectly modulated source (e.g., MZM, IQM). Recently, chip-based frequency comb sources were demonstrated for C-band WDM data transmission, such as a 12 Tbps transmission based on a mode-locked laser[47]. In long-haul transmission, the cost is dominated by the fiber installation; hence, maximizing the spectral efficiency is crucial to optimize the cost per bit. Accordingly, QD-MLLs are not envisioned as a viable solution for that application use case, where the optical bandwidth and the power per line is the determining factor. On the contrary, intra-DCIs are fiber-rich environments and each optical fiber typically transmits data on 4 carriers with 20 nm spacing, to avoid chromatic dispersion and employing uncooled lasers. For these reasons, QD-MLLs offer a better alternative when used with coherent OFTS over the intra-DC transmission reach for the following reasons: (i) they have high power conversion efficiency, because the optical power is concentrated in a limited optical bandwidth unlike soliton-based combs; (ii) they are not totally uncooled, but a single TEC can tune n-λs; and (iii) employed in coherent OFTS, the chromatic dispersion and fiber non-linearities penalties are minimized (see "Non-linearities in comb laser based OFTS"). Thus, comb based OFTS are fit for short reach transmission since they typically employ a limited number of lanes per fiber. This contrasts longer haul where the lack of

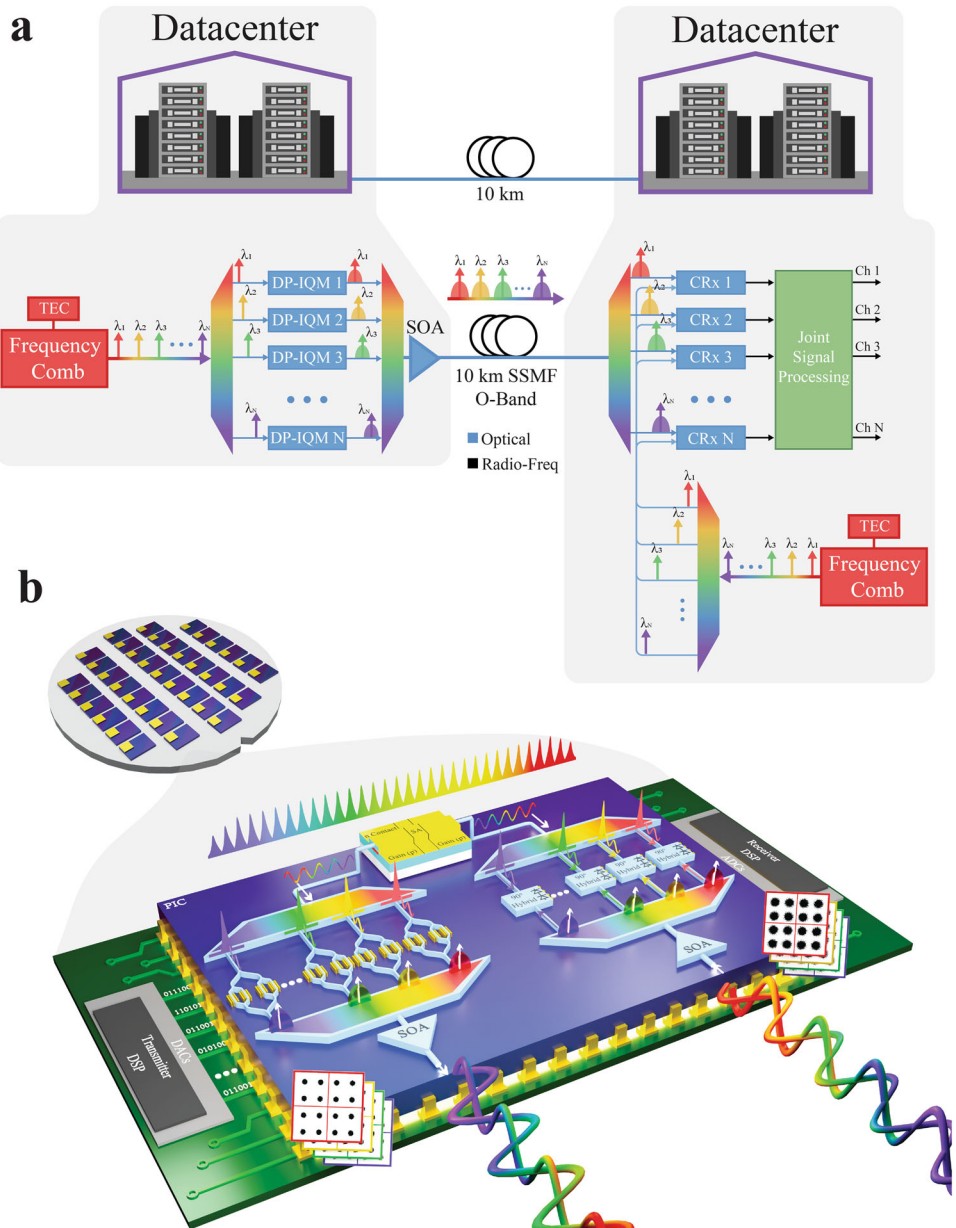

**Fig. 2 | Coherent comb-based links. a** Schematic of comb-based DCI 10 km links using two matched and independent combs, one each for carrier and LO. **b** Concept of a comb-to-comb coherent TROSA consists of a series of DACs transmitting *n*- channels modulated by n-DP-IQ modulators driven by one QD-MLL. The comb source is also used as the Local Oscillator (LO) used to receive *n*-channels of data using *n*-coherent receivers.

fiber requires operation over larger bandwidths (e.g. C + L bands). These use cases could favor microresonator-based solitons comb sources. These can generate hundreds of comb lines over very large bandwidths[18,28]. However, the non-uniformity of their output can often lead added complexity for some channels within the desired bandwidth. Additionally, the temperature sensitivity of the microring resonators used can also be an issue.

QD-MLLs emit high frequency pulses with a fixed spacing determined by their cavity length. They are compact, efficient, and robust and do not require initiation routines or feedback loops[48]. Their efficiency can also be optimized by controlling the length of their saturation absorber (SA)[48]. For example, a 20-line comb configuration would have a single sided wall plug efficiency (WPE) of around 7% if the SA length was 10% of the cavity length[48]. For comparison, soliton combs have been demonstrated with 40–66% efficiencies[15,49], however this often does not include the pump laser power, which should also be

included in the net WPE calculation for the source, resulting in less efficient combs. QD-MLLs are also very stable. For this paper, the comb sources were operated for a maximum of 30 continuous hours with no measurable impact on system performance. The phase noise of the QD-MLL has also been previously studied[50] and our results validate its properties as the comb can be used for both carriers and LOs (see "QD-MLL as local oscillator").

In this work the QD-MLLs were characterized to optimize the transmission performance by changing the driving current, the SA bias voltage, and the temperature. Figure 3a–c shows the linewidth, power per line, and number of lines within 3 dB, respectively, attainable by varying the comb state at a constant temperature of 42 °C. Figure 3d shows the optical spectra of the carrier laser (red) and LO laser (blue) optical spectra. Increasing the SA bias increased the number of lines within 3 dB of the max line but decreased the maximum power per line. This could be mitigated by increasing the driving current. The

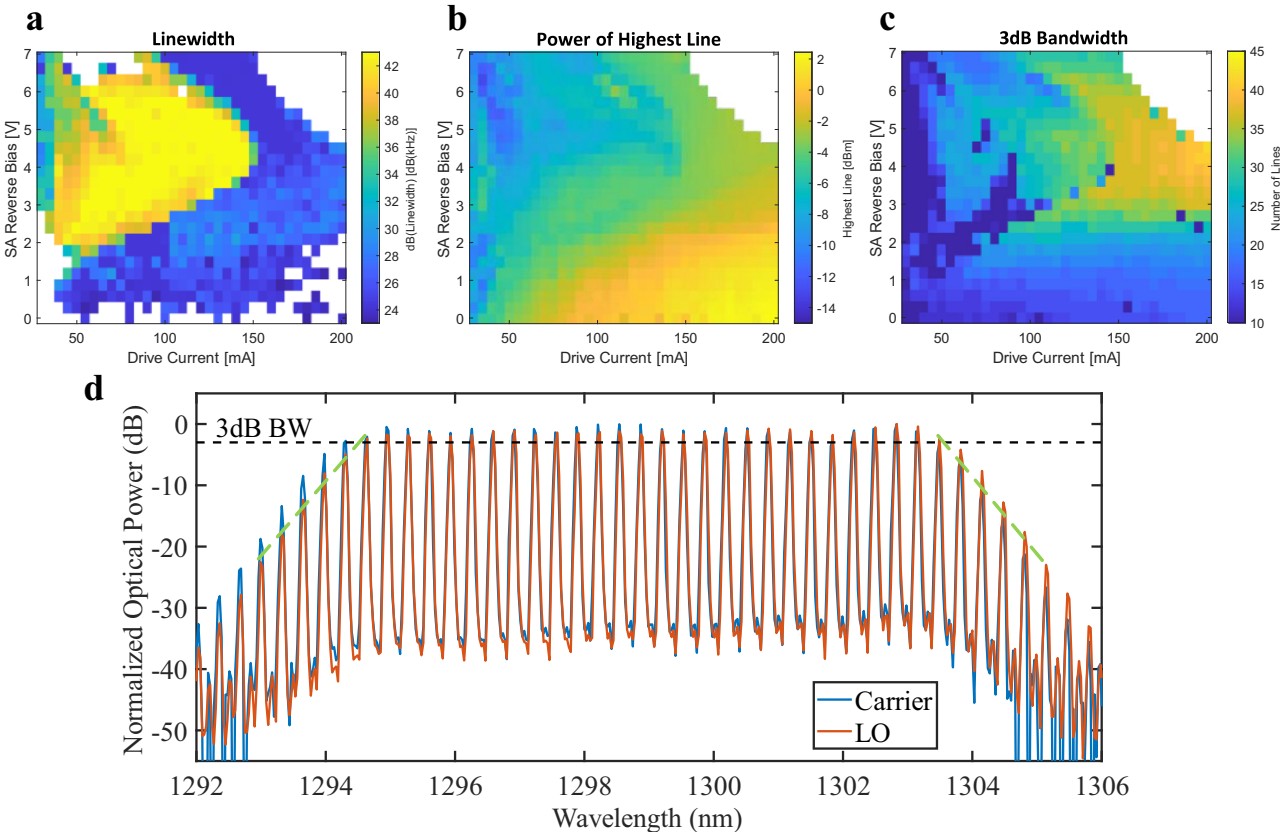

**Fig. 3 | QD-MLLs characterization.** Different comb states were characterized and compared through their (**a**) linewidth, (**b**) power per line, and (**c**) number of lines within 3 dB of the highest-powered line for different drive currents and saturation absorbers (SA) bias voltages. **d** Optical spectra of both combs. Diagonal green dotted lines show a 12.5 dB/nm output power roll off for out of band λs not used in transmission.

temperature primarily affected the central wavelength. The spacing of the lines was determined by the cavity length and was fixed at 58.2 GHz. The center wavelength was set at 1299 nm and had a -10 kHz linewidth. The comb was operating at 160 mA, 2.2 V, and 42 °C and produced 28 lines whose optical powers were within 3 dB of each other (per Fig. 3d). The min power per line was −3 dBm within the desired bandwidth (BW). The low phase-noise of the QD-MLL is ideal for use as a LO. Additional information about the QD-MLL used can be found in Supplementary Information S4. Experiments also show that QD-MLL performance is comparable to an ECL and -0.7 dB better than a DFB at the same received optical power (ROP) as further detailed in "QD-MLL as local oscillator".

Here we define the power roll-off slope for a comb laser (operating as a carrier and/or an LO) as the Rate of Side Mode Suppression (RSMS) with units of dB/nm and applying to λs that are not modulated and/or used as an LO. As seen in Fig. 3d, to the left and right of λ1 and λ28, respectively, there is a 12.5 dB/nm RSMS for out of band λs not used in transmission. In general, as the RSMS increases, the efficient use of the modulated/received λs increases. This translates into more efficient use of the power consumed when operating the comb laser. For example, quantum-well and quantum-dash MLLs have an RSMS of -8.92 dB/nm and -dB/nm[51]. These combs are less efficient but still suitable for OFTS, as opposed to sources such as integrated electro-optic frequency combs which have a RSMS values of -0.54 dB/nm[52], -0.28 dB/nm[18], and 0.50 dB/nm[53]. These low RSMS values translate into multiple unused λs being generated.

**10 km transmission**

Fig. 4 summarizes the comb-to-comb coherent OFTS experimental results. The BER for each wavelength was tested using QD-MLLs as

carrier and LO over 10 km of SMF operating at 56 GBaud and differing modulation formats. All 28 lines were tested in a WDM configuration using a sliding window and bulk modulation. Specifically, we used a Tunable Bandpass Filter (TBF) at the transmitter with a fixed bandwidth of $\Delta\lambda = 290$ GHz such that 5 comb lines were bulk modulated simultaneously—two aggressors on either side of the central modulated wavelength. This value was chosen such that the optical power throughout of the experimental setup would not saturate the PDFAs in the absence of low-loss O-band optical Mux/DeMux pairs at the desired comb spacing. At the receiver, the desired LO λ is passed through a TBF and delivered to the coherent receiver. The BER of each line is shown in Fig. 4a and summarized in the Supplementary Table S2. We achieved net 10.15 Tbps using 26 lines at 56 Gbaud DP-16QAM assuming the 14.8% overhead O-FEC. We also achieved 12.14 Tbps using 26 lines at 56 Gbaud DP-32QAM assuming the 20% overhead SD-FEC. The resulting 16 and 32 QAM constellations at 1296, 1298, 1300, and 1302 nm are shown in Fig. 4b. Figure 4c shows the resulting optical spectrum of the signal sliding window with the corresponding LO (orange) and all modulated channels (blue) with LOs superimposed.

The low drift noise of the comb lasers allowed for the carrier and LO combs to be set during the experimental process and subsequently not adjusted during of transmission capacity characterizations. The results were primarily limited by the power per line penalties coming from the saturation of our PDFAs while amplifying 5 channels at once. The characteristics of the QD-MLL as a carrier in a coherent transmitter were examined in "QD-MLL as signal carrier". Also, small linear cross talk penalties are detailed in Supplementary Information S5. Overall, these results suggest that adding or removing more comb lines is possible without incurring significant penalties, thus making different comb source configurations a possibility.

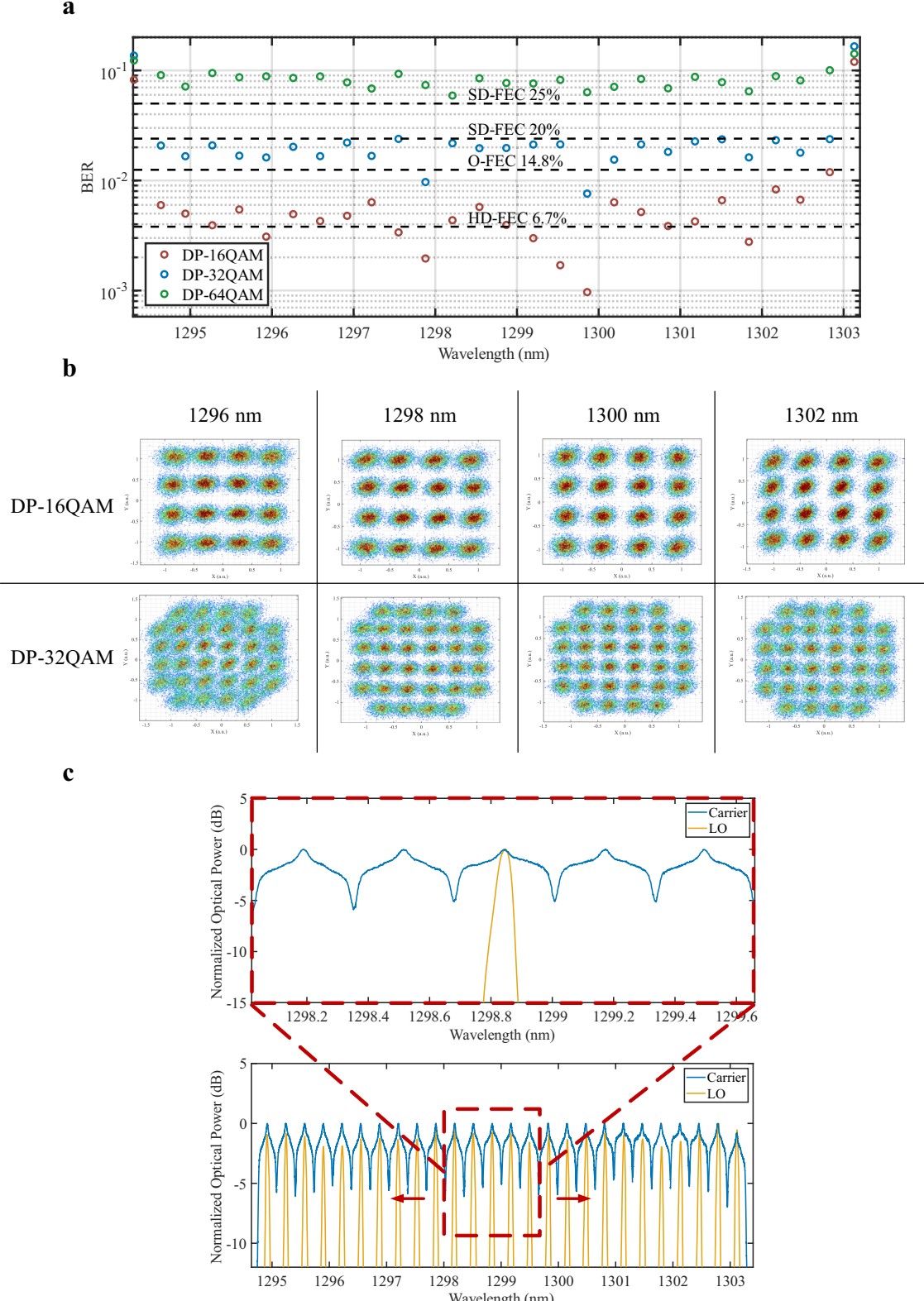

**Fig. 4 | Comb-to-comb coherent OFTS experimental results. a** Bit Error Rate (BER) for each line tested using a TBF sliding window using two QD-MLL as carrier and LO after 10 km SMF operating at 56 GBaud for different modulation formats. **b** Resulting 16 and 32 QAM constellations at 1296, 1298, 1300, and 1302 nm. **c** Resulting optical spectrum of signal sliding window with corresponding LO (top) and all modulated channels with LOs superimposed.

All 28 lines were also tested using a sliding window in an IMDD configuration. The bandwidth of the TBF is kept constant such that 3 comb lines are bulk modulated at 56 Gbaud, and the central line is optimized to mimic a WDM scenario with 2 aggressor channels. The

penalties of bulk modulating are presented in Supplementary Information 8 Similarly to the coherent case, this value allows for not stressing the optical power budget of the experimental setup. The fiber length was set to 10 km and the ROP was kept constant at 2 dBm. The

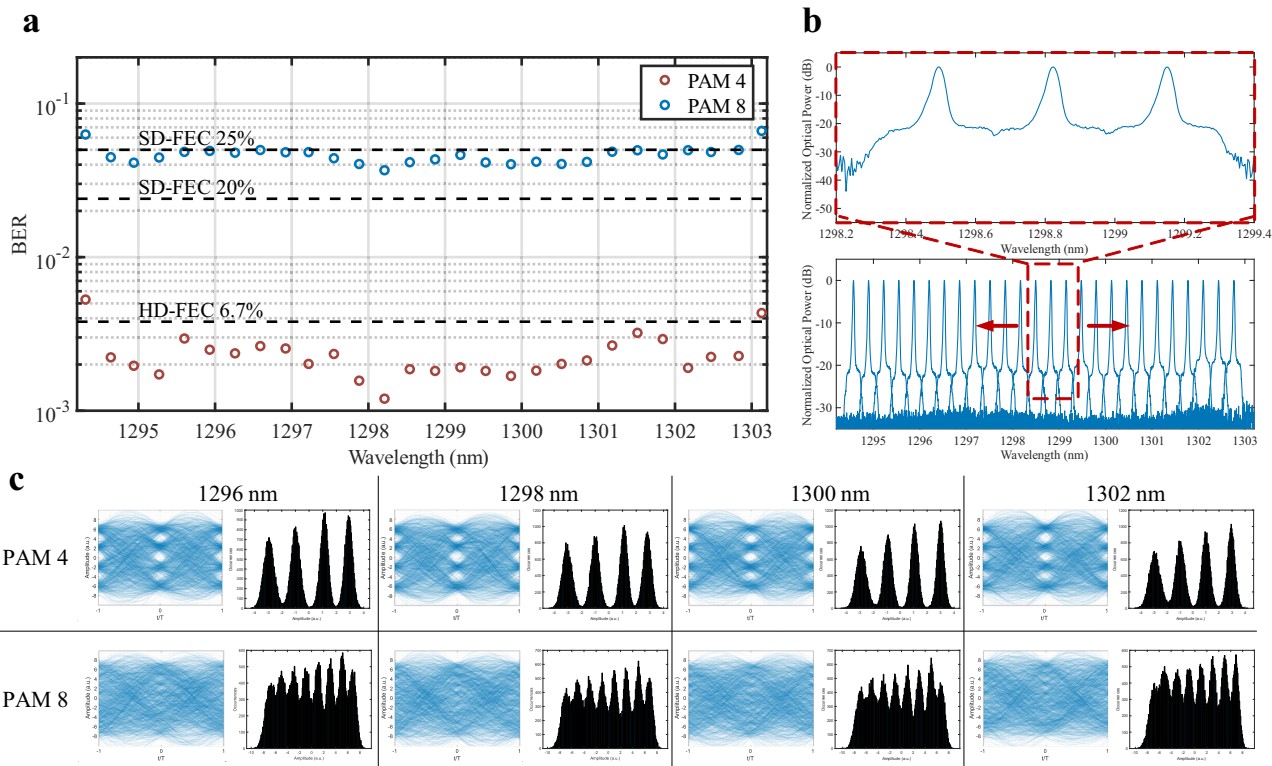

**Fig. 5 | Comb IMDD OFTS Experimental results. a** Bit Error Rate (BER) for each line tested using a TBF sliding window after 10 km SMF operating at 56 GBaud for different modulation formats. **b** Resulting signal spectrum of sliding window (red dashed line). **c** Resulting PAM 4 and 8 eyes and histograms at 1296, 1298, 1300, and 1302 nm.

same comb state was used operating with 160 mA driving current, 2.4 V saturation absorber voltage, and 42 °C. Figure 5a shows the BER achieved for each comb line using different PAM orders and FEC thresholds. We achieved net 2.92 Tbps using 28 lines at 56 Gbaud PAM4 assuming the 6.7% overhead HD-FEC for the inner 26 channels and the 20% SD-FEC for the 2 extremities channels. We also achieved 3.49 Tbps using 26 lines at 56 Gbaud PAM8 assuming the 25% overhead SD-FEC. The decrease in power per line of the outer channels causes an increase in the BER penalty and therefore limits the number of functioning comb lines for this comb state. Our findings confirm the usage of QD-MLL for O-band, 10 km, IMDD systems at the expense of a lower system capacity with respect to its coherent counterpart, as explained below.

Supplementary Table S2 shows the summary comparison between IMDD and coherent comb based OFTS for different modulation formats. For a fair comparison, we compared the different OFTS using a single FEC threshold per configuration. The coherent DP-16QAM results show a 3.72-fold increase in net capacity from the PAM 4 comb results over 10 km in the O-band. The IMDD performance and sensitivity compared to coherent systems is further examined in "Nonlinearities in comb laser based OFTS" and Supplementary Fig. S5.

## Nonlinearities in comb laser-based OFTS

The nonlinear degradation of the systems was numerically investigated by increasing the total average input power into the fiber for different numbers of comb lines. The SNR penalty of both systems in the O-band is presented in Fig. 6a. Solid lines correspond to the coherent calculation, while dashed lines were assigned to the IMDD results. Note that the x-axis determines the total input power value irrespective of the number of lines launched into the fiber, hence a lower number of lines correspond to a higher input power per line. The most evident observation from the figure is that the onset of nonlinear penalty in the coherent case is always found at a higher power with respect to the corresponding IMDD case; in agreement with the

literature[54]. This means that comb-based coherent OFTS are less sensitive to Kerr nonlinear impairments when compared to equivalent IMDD OFTS operating at the same baud rate and over the same distance. This finding favors the use of coherent modulation in comb-based systems. The figure also shows that, in contrast to the IMDD case, the SNR penalty of the coherent OFTS exhibits low variations when five or more channels are launched into the fiber. The SNR is mainly mediated by the total input power instead. In the IMDD case, a decrease in the number of comb lines reduces the nonlinear penalty. However, even by reducing the line count to five, the performance is worse when compared to 25-line coherent OFTS. This means that the performance of the coherent system will always be superior in terms of nonlinear penalties. The variation of the line spacing from 58 to 65 GHz for the 25-line simulations led to practically no variation in the SNR penalty for the analyzed situation due to the Mux and DeMux brick wall bandwidths, preventing any noise from neighboring channels from affecting the signal. Further penalties of these components are analyzed in Supplementary Information S5.

Greater insight can be gained by analyzing the nature of the nonlinear behavior for each modulation format. In the case of PAM4, the single-line simulation indicates that self-phase modulation (SPM) does not affect the performance even at +13 dBm of total input power. This is expected due to the absence of chromatic dispersion in the O-band, thus disabling the necessary phase-to-amplitude modulation conversion that is responsible for SNR degradation. Therefore, the nonlinear cross-talk (XT) is mainly mediated by four-wave mixing (FWM) in the PAM4 case. The predominance of FWM is an expected result due to the very low value of chromatic dispersion exhibited by the optical fiber in the O-band and the uniform spectral separation between the comb λs for the considered channel plan. In contrast, the coherent transmission case where data is encoded into the optical waveform phase, SPM causes degradation of the performance for high input power in O and C-band cases. This is observed in the single-channel coherent transmission simulation. When multiple comb lines

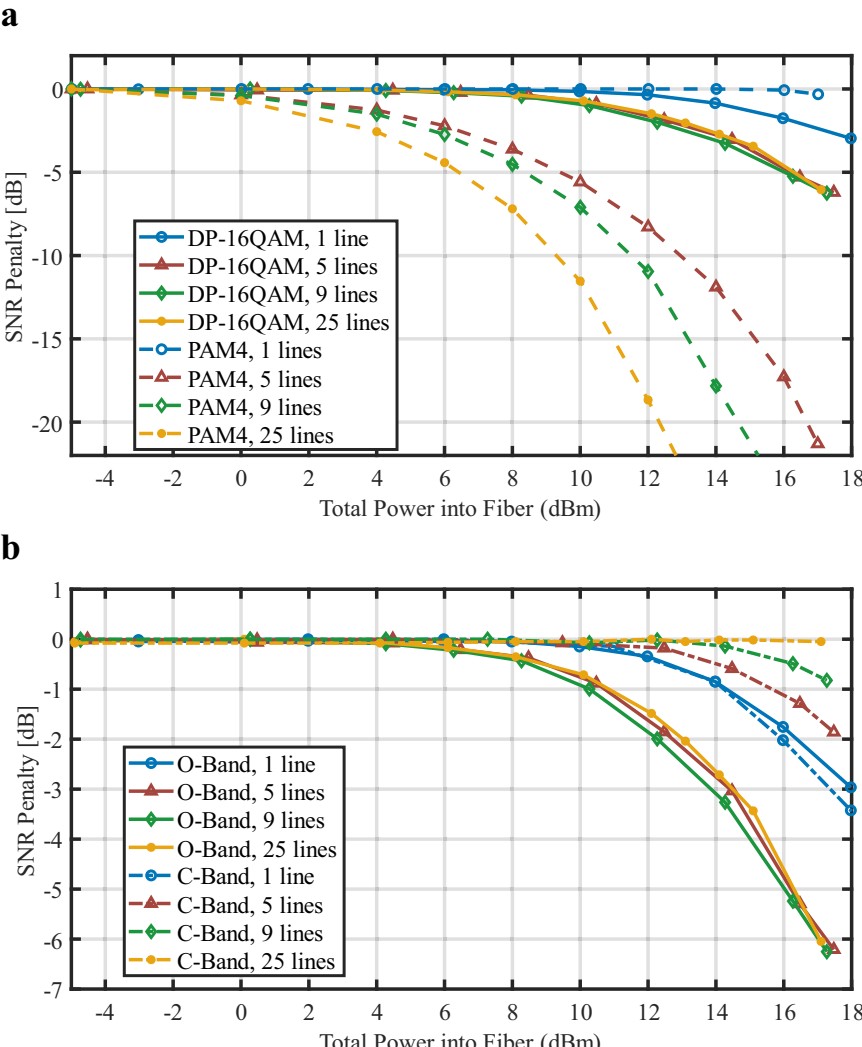

**Fig. 6 | Nonlinear simulations.** Calculated SNR penalty as a function of total input power into the fiber for (**a**) PAM4 (dashed) and DP-16QAM (solid) modulation formats running at 56 GBaud in the O-band and (**b**) coherent C-band and O-band 10 km OFTS. A comb with variable number of lines was used as a transmitter and as the LO when applicable.

are injected, XPM will further degrade the overall performance of the system in the O-band analysis. The very low penalty variation as a function of line count and line spacing in the coherent system analysis points out that phase modulation is responsible for the nonlinear penalty rather than FWM. These differences stem from the fact that in the IMDD case the optical carrier is not suppressed; hence, it propagates through the fiber activating several non-linear processes, especially FWM. To reduce FWM in IMDD DWDM systems, multiple solutions have been proposed such as using unequal channel spacings[55] or using orthogonal polarization launch of neighboring channels[56]. FWM penalties will be impacted by PMD because this leads to depolarization along the transmission line. As PMD increases, the FWM penalty is reduced and forms a distribution due to the randomness of total differential group delay (DGD) and the orientation of the principal state of the polarization vector relative to the launch polarization[57]. Note that we indeed included PMD in our simulations. When only four channels are launched the use of a YXXY polarization at the input decreases the FWM impairment. This has been proposed to implement the 800G-LR4 Ethernet node. This can be generalized to 8 channels as XYYXXYYX or to 26 channels as shown in Supplementary Information 7. On the other hand, coherent transmission requires suppressing the optical carrier, thus improving its tolerance to fiber non-linearity significantly as detailed earlier.

The difference between coherent transmission in the O-band and the C-band as a function of total power vs. SNR was also calculated as seen in Fig. 6b. Using the same simulation setup as above, the non-linear penalty for coherent OFTS operating on both bands was compared. We observed that when the number of channels is increased and thus the power per line is reduced, the nonlinear penalties, mainly those mediated by phase-modulation (as discussed above), also become reduced in the C-band. This is associated with the symbol sequence walk-off induced by fiber dispersion present in the C-band that reduces the efficiency of the multi-line nonlinear processes, and the reduction of the power per line as the line count increases, thus reducing SPM. Both scenarios suggest that the total power into the fiber should be kept below 6 dBm, irrespective of the employed band, to avoid any major nonlinear penalties in comb based coherent OFTS.

For the simulations in this paper, we only considered the nonlinear (NL) effects at 10 km as this was the use case of our study. We would expect the NL effects to diminish for distances >10 km because the light power traveling in the fiber reduces over distance. This effect is magnified by the higher attenuation in the O-band as compared to the C-band as can be seen in Fig. 6b. The limit for our use case of coherent transmission in the O-band over 10 km is -6 dBm of total input power into the fiber. The resulting SNR penalty from the NL effects induced at this power cause an increase in the BER over the

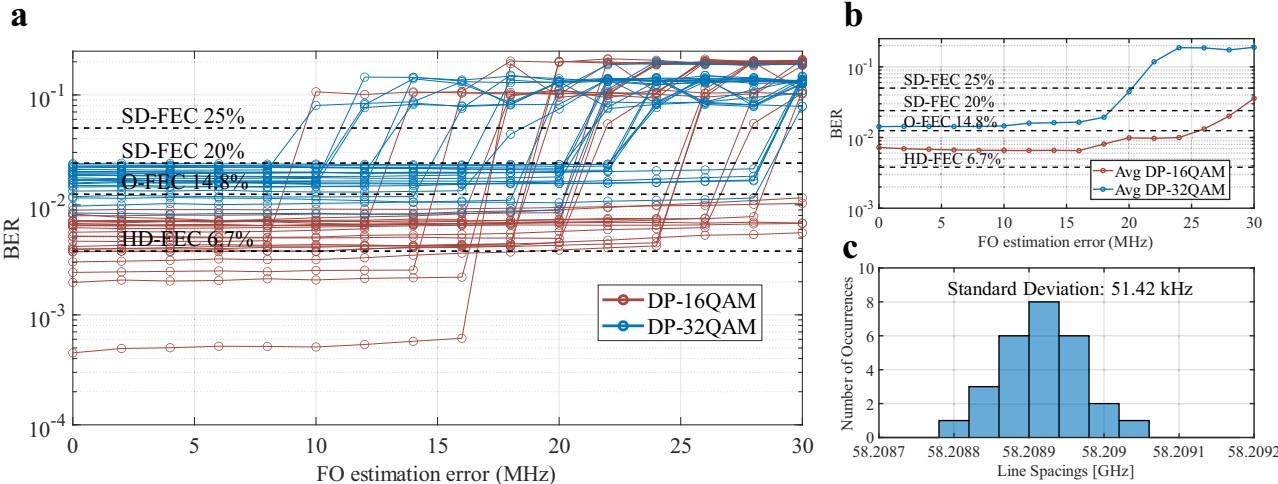

**Fig. 7 | FO estimation error calculation. a** Bit Error Rate (BER) for one channel using an FO estimation offset for a comb-to-comb coherent O-band transmission for both DP-16QAM and DP-32QAM over 10 km. **b** Resulting average BER from both sets of 26 lines using the wrong FO correction. **c** Histogram of average channel spacing variations based on 270 measurements.

targeted FEC overheads. This would result in a throughput reduction from 10.18 Tbps to 9.71 Tbps and from 12.14 Tbps–11.65 Tbps for the DP-16QAM and DP-32QAM results, respectively.

### QD-MLL as local oscillator

The efficiency of a second comb source acting as LO was experimentally demonstrated by comparing its performance in an O-band OFTS against other lasers. We measured the optical power penalty of a coherent system employing an external cavity laser (ECL), a distributed feedback laser (DFB), and another almost identical comb. A setup similar to the setup used for the coherent transmission was used with interchangeable LOs. The LO consists of either a tuneable ECL, an array of DFBs, or a second QD-MLL. The ECL and DFBs both had an output power of 15 dBm, a linewidth of ~500 kHz and 1 MHz, and a side-mode suppression ratio (SMSR) of 45 and 50 dB, respectively. The second QD-MLL was connected to an EXFO XTM-50 narrow TBF and an 18 dB PDFA to select a single comb line. The same comb state was used for both lasers. The bandwidth of the Santec TBF is kept constant such that 3 comb lines are bulk modulated at 56 Gbaud, and the central line is optimized to mimic a DWDM scenario with 2 aggressor channels. The fiber length was kept at 10 km. The ROP at the balanced PDs was kept constant at +2 dBm using a constant LO power of +15 dBm. The measured BER between the 3 cases for a DP-16QAM signal was minimal. Both the ECL and the comb source obtained a BER of 0.0026 while the DFB achieved 0.0033 at the same ROP. By increasing the optical power of the DFB by 0.7 dB, the same BER was achieved for all three cases. This suggests that the individual comb lines are equally as effective as an ECL when acting as an LO. Additionally, the 0.7 dB optical power penalty for using a DFB as an LO in the O-band further enhances the claim that O-band short-reach coherent DWDM systems support next-generation DCI architectures. Our results further the claim that coherent comb to comb configurations can be used for high speed OFTSs. Given the right laser noise parameters and power, this type of coherent configuration could be implemented independently of the type of comb source technology used.

The penalty for applying a shifted FO correction factor using the corresponding DSP algorithm on the different channels was measured. This analysis is relevant to ensure that the different frequency offsets produced from the drift and channel spacings of either the carrier or the LO comb could be compensated using DSP techniques. Figure 7a shows the applied FO estimation offset penalty on transmission performance. The measured offset was applied to the transmission results

for all DP-16QAM and DP-32QAM data captures. This additional offset of 1–30 MHz was applied to the results and the resulting BER was calculated. Our results show that the equalizer can on average compensate for an offset up to 22.1 MHz and 19.2 MHz for the DP-16QAM and DP-32QAM cases, respectively as seen in Fig. 7b. This sensitivity is enough to cover the comb laser frequency drift, which is in the kHz range[58], and the comb λ spacing variations. The mean line spacing was 58.2 GHz with a standard variation of 51.42 kHz, orders of magnitude below the recoverable FO estimation error range. These experimental results validate the use of a single FO estimation step when determining the FO correction needed for every line in the comb.

### QD-MLL as signal carrier

The transmission performance of a single comb line as the carrier over 10 km using different symbol rates can be seen in Fig. 8. Using a DFB as LO, we achieved net 1.6 Tbps using 167 Gbaud DP-64QAM under the 25% overhead SD-FEC threshold. Additionally, we achieved net 1.13 Tbps using 136 Gbaud DP-32QAM under the 20% overhead SD-FEC threshold and net 1.09 Tbps using 156 Gbaud DP-16QAM under the 14.8% overhead O-FEC threshold. These results prove the benefits of using QD-MLLs because they can be used to transmit using higher constellation cardinalities such as 64QAM due to their low phase noise and small linewidth. These results are limited by the 70 GHz bandwidth of the O-band BPDs used in the setup. The symbol rate of comb based DWDM systems is limited by the channel spacing. The signal bandwidth cannot be larger than the spacing between comb lines due to the penalty of linear cross talk between WDM channels (see Supplementary Information S5). However, these results demonstrate the potential for O-band QD-MLL to support higher symbol rates given larger comb spacings.

### Discussion

This work presents a comparison of potential solutions for increasing DCI capacities whilst reducing power consumption, complexity, and cost. The work demonstrates the ways and means to build wide (e.g., 20 lanes), fast (e.g., 400 Gbps/λ) systems (e.g., 12 Tbps) versus very narrow (e.g., 1 λ) high capacity (e.g., 1.6 Tbps) or extremely wide (40+ lanes) and very low capacity per λ (e.g., 100 Gbps/λ) O-band OFTS for <10 km SMF reaches.

In the proposed scheme, every line does not require its own amplifier. It will benefit from recent developments of quantum-dot semiconductor optical amplifier (QD-SOA). These QD-SOAs have been

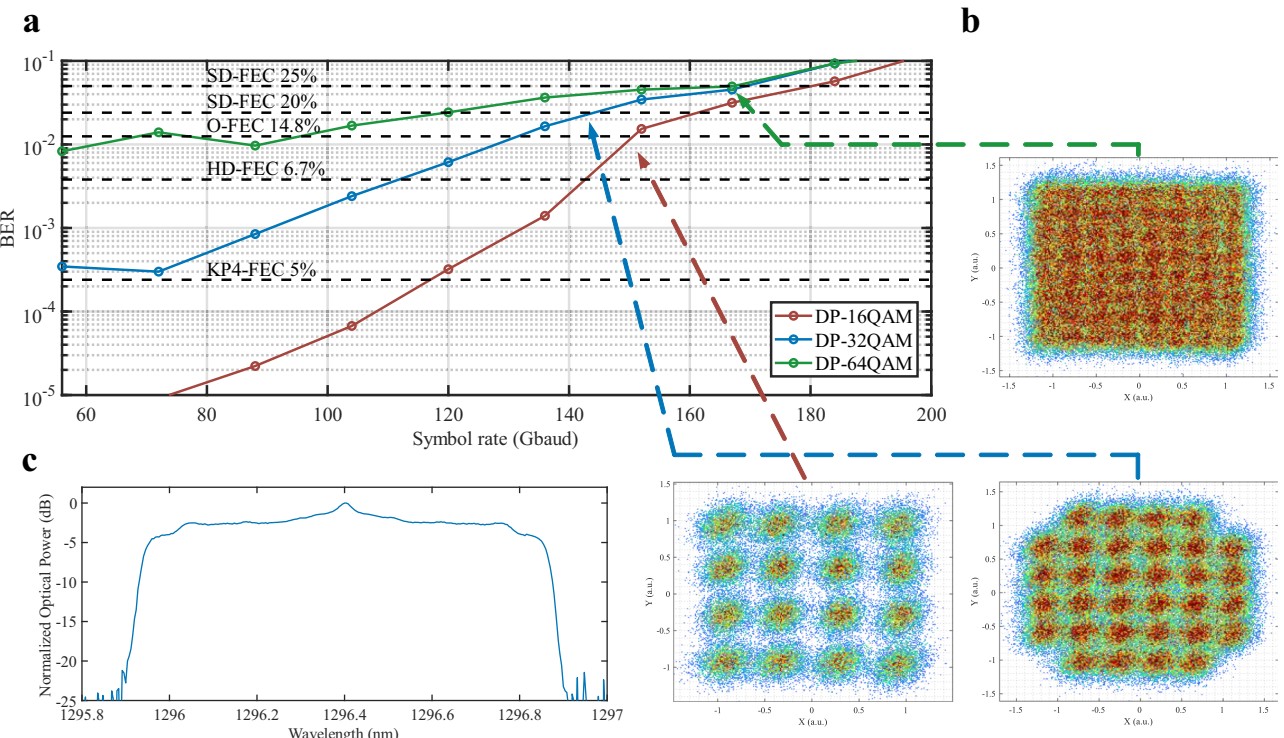

**Fig. 8 | Single comb line coherent OFTS Experimental results. a** BER of a single comb line after 10 km using different symbol rates at DP-16, 32, and 64QAM. **b** Resulting DP-16QAM, DP-32 QAM, and DP-64QAM constellations at 152, 136, and 168 Gbaud, respectively. **c** Resulting optical signal spectrum of 1.6 Tbps net capacity at DP-64QAM.

demonstrated capable of supporting high speed WDM coherent systems[34]. In addition, a single SOA can be used to amplify 4 lines at once[34]. In this sense, our scheme does not require 26 SOAs. Our analysis shows that 7 amplifiers can be used to operate the concept presented in Fig. 2.

In order to compare the power consumption of individual laser sources to a comb-based system with SOAs, the following assumptions were made, (i) both systems would require the same optical link budget per λ, (ii) one SOA was needed for every 4 comb channels, and (iii) each individual laser source would consist of a DFB and requires a TEC. The typical values used for the DFBs were a slope efficiency of 0.29 mW/mA, a bias voltage of 1.6 V, and a threshold current of 15 mA[59]. Following the power consumption comparison established in ref. 44, the total link budget for a 10km O-band WDM coherent system with 26 channels would be around 17.4 dBm. This would result in an assumed laser power of 6.2 dBm. Each TEC is assumed to consume around 1.2 W[60].

Operating 26 DFBs with 6.2 dBm each would consume an aggregate of 0.1658 W × 26 lasers = 4.31 W. Each laser source would need its own TEC, resulting in 26 TECs total for an additional 26 × 1.2 W = 31.2 W. The total power consumption from the laser sources would be around 35.51 W. On the other hand, a single QD-MLL consumes around 352 mW. A QD-SOA amplifying 4 λs simultaneously would consume an additional 1.04 W to obtain the same power per channel as the individual lasers. This would result in a total power of 0.352 W + 1.04 W × 26/4 = 7.632 W. The number of TECs is also reduced from 26 to 8 as the comb-based system only requires one TEC for all laser channels and 7 for the SOAs. This would result in an estimated TEC power consumption of 7 × 1.2 W = 8.4 W and a total laser source power of 16.03 W. This represents a 75% reduction in TEC power consumption. This results in a total power consumption reduction of 45.1% when switching to a comb based OFTS as opposed to single-carrier laser sources.

Additionally, through design, analysis, and experimentation, we quantify the optimum comb line spacing for DWDM DCI OFTS

deployments. Specifically, the proposed use case does not need an abundance of λs, which supports implementing QD-MLL combs. In contrast, many C-band Kerr MLL OFTS demonstrations have been reported to date implemented using many λs, often inefficiently. Because of the DCIs constraints, only a modest number of λs are required to scale the capacity from the current 400 Gbps capacities to 12.8 Tbps. Also, saturable absorber based QD-MLL combs provide the optimum number of λs, with enough power per λ, and the optimum spacing to enable optimizing Gbaud/λ, and therefore overall high OFTS capacities.

A summary of the trade-offs for designing solutions that use differing numbers of comb channels and spacings can be found in Table 1. For this comparison, the following assumptions were made: (i) the optical bandwidth of the comb-source is fixed to 1.2 THz, and (ii) all the scenarios achieve the same capacity. The optical bandwidth is limited by the type of comb source structure used. Therefore, increasing the number of channels can only be done by reducing the spacing between the comb tones. Additionally, by using DWDM transmission configurations, we expect that all the scenarios can achieve the same aggregate net rate by using the same modulation format since the product of baud rate and number of lines is fixed. This comparison is not limited to QD-MLL as the other comb sources can achieve similar spacings and will therefore have similar required RF bandwidths and standard compatibilities.

Having many lines is impractical due to the large number of components needed to build the system. Additionally, these systems will have a very large optical power budget penalty due to the output power of the comb source being distributed among multiple lines. The high number of channels will also lead to a high nonlinearity penalty due to the aggregate high total power required to operate so many links at once. The SNR penalty of adding more lines to the system can be seen in Supplementary Information S6. However, these systems have the advantage of only requiring low performance components to operate their relatively small symbol rates. For example, a 40-channel comb source-based system would only need ~15 GHz of electro-optic

**Table 1 | 10 km, O-band, coherent optical fiber transmission system design, and attendant comb laser carrier and LO performance comparison**

| # Lines | Baudrate (Gbaud) ≡ Δν Line spacing (GHz) | Req. RF / electro-optic band-width (GHz) | Req. CMOS node for the ASIC engine | Optical power budget pen-alty (dB) | Non-linearity penalty (dB) | Projected power con-sumption per bit (W/Gbps) | Standard compatibility / output granularity |
|---|---|---|---|---|---|---|---|
| 40 | 30 | 15 | 28/14/10 nm | -16.8 | 3.65 dB | ~0.06 | 100 / 200 G |
| *26* | *56* | *28* | *NA* | *-14.1* | *3.1 dB* | *~0.05* | *This Work* |
| 20 | 60 | 30 | 10/7 nm | -13.8 | 2.8 dB | ~0.045 | 400 G |
| 12 | 100 | 50 | 7/5 nm | -10.8 | 1.37 dB | ~0.0325 | 600 G |
| 8 | 150 | 75 | 5/3 nm | -9 | 0.58 dB | ~0.0275 | 800 G |
| 6 | 200 | 100 | 3 nm | -7.8 | 0.27 dB | ~0.021 | 1.6 T |
| 4 | 300 | 150 | 2 nm | -6 | ~ 0 | ~0.018 | 2.4 T |

*italics* this work

bandwidth which is compatible with standard 100/200 G systems and can be built using cost effective 14/10 nm CMOS nodes. Older CMOS nodes, however, increase the power consumption of the system. Having a slightly larger channel spacing of 60 GHz with 20 comb lines still requires many components which can be difficult to integrate. However, the compatibility with existing 400ZR solutions could support the deployment of this configuration at scale. A 100 GHz spacing solution still supports deployment using current CMOS nodes and will not have a large power penalty. However, the output granularity of 600 G is uncommon, which hinders its scalable deployment. As the channel spacing is increased, the number of channels decreases and so the optical penalty power budget penalty also decreases. In turn, this will reduce the nonlinearity penalties. Additionally, the need for smaller CMOS nodes to provide the required electro-optical bandwidth will result in a more efficient configuration. A comb source with 150 GHz spacing would have a low nonlinearity penalty and would be compatible with 800ZR solutions. Going to larger spacings would result in very expensive CMOS nodes such as 3 nm or even 2 nm and with electro-optic BWs that would be very challenging to produce in scale. These nodes and the maturity of the current 1.6 T+ solutions will make these configurations more expensive to deploy at scale. This comparison can be applicable to various comb source technologies, however a deeper understanding of the fabrication limitations of QD-MLL comb sources provides further insight into their potential use in multi-Tbps OFTS.

A summary of the trade-offs for building QD-MLL with different channel spacings and therefore number of comb channels can be found in Table 2. For this comparison, the following assumptions were made: (i) the optical bandwidth of the comb-source is fixed to 1.2 THz, (ii) the comb-based system will operate using DP-32QAM, (iii) the QD-MLL is built with 6 quantum dot layers, (iv) the QD-MLL operates with second harmonic operation, and (v) operating temperatures between 20 to 60 °C. The optical bandwidth of the sources is assumed constant, and the same modulation format is used. The construction of the comb laser is assumed equal such that the laser properties (e.g. linewidth, SMSR) are the same for all cases. A second harmonic operation is assumed to create larger comb spacings. The cavity length is calculated assuming Fabry Perot formula and an InP index of refraction of 3.2[61].

A line spacing lower than 25 GHz does not require 2nd harmonic to be viable, however it imposes the need for too many lines to achieve the same net capacity. This many optoelectronic and electrical components are not feasible to integrate. Utilizing the 2nd harmonic of the cavity resonance, efficient comb laser sources can be built with channel spacings ranging from 50 to 100 GHz. These sources have enough gain and provide enough output power to be used for an OFTS. For spacings of 150 GHz and above, the cavity length becomes too small to be practical for quantum dots. The output power also diminishes due to gain saturation. Higher order harmonic mode locking is feasible to

**Table 2 | QD-MLL designs for 10 km, O-band, coherent optical fiber transmission system design feasibility and comparison**

| # Lines | Δν Line spacing (GHz) | Cavity length (mm) | Baudrate (GBaud) | Power per line (dBm) | Total output power (dBm) | Max capacity @ DP-32QAM (Tbps) |
|---|---|---|---|---|---|---|
| 48 | 25 | 1.88 | 25 | -6 | 10 | 12 |
| 24 | 50 | 1.88 | 50 | -3 | 10 | 12 |
| *26* | *58.5* | *1.35* | *56* | *-3* | *10* | *12.14* |
| 12 | 100 | 0.94 | 100 | 0 | 10 | 12 |
| 8 | 150 | 0.63 | 150 | 0 | 8 | 12 |
| 6 | 200 | 0.47 | 200 | NEG | NEG | 12 |
| 4 | 250 | 0.38 | 200 | NEG | NEG | 8 |

*NEG* Not Enough Gain, *italics* this work

obtain these higher line spacings, however this would diminish the efficiency of the QD-MLLs and hence have been omitted from this comparison. Cavity lengths above 200 GHz would require cavity sizes below 0.5 mm. These cavities would not provide enough gain to generate a functioning QD-MLL as the mirror losses would be too high. Additionally, having greater line spacings (250 GHz and above) would not provide any additional gains as current systems cannot operate over 200 Gbaud and the number of lines is significantly reduced. This would result in less lines operating at the same rate for diminishing returns compared to systems with 200 GHz spacings. Some of the limitations of QD-MLLs comb sources with high spacings can be overcome by using different comb technologies. For example, designa with spacings >100 GHz can be generated using microresonator-based solitons[18]. However, the specific use case must be carefully examined before selecting one technology over another, with parameters such as efficiency, RSMS, power per line, stability, phase noise being key factors to OFTS performances.

The ideal spacing for comb based coherent OFTS deployed in short reach applications of 10 km or less depends on many factors such as the optical power needed, the resulting nonlinear penalties, the number of components needed, the comb laser fabrication feasibility, and ultimately the power consumption of the entire system.

In conclusion, we present an O-band coherent optical fiber transmission system based on QD-MLLs using two independent free running comb lasers, one each for the carrier and the Local Oscillator (LO). Using a comb-to-comb configuration, we demonstrate a 10 km single mode fiber, O-band, coherent, heterodyne, 12.1 Tbps system operating at 0.47 Tbps/λ using 26 λs. This capacity represents a 3-fold increase in comparable QDash-MLL demonstrations, built using comb-to-comb coherent transmission systems. Through design, analysis, and experimentation, we detail the merits of operating in the O-band and using coherent transmission for the DCI network segment. By

## Methods

### Detailed coherent experimental setup

Supplementary Figure S17 shows the coherent experimental system. The transmitter (Tx) source consisted of a QD-MLL connected to an O-band Santec OTF-350 optical tuneable bandpass filter (TBF) with a bandwidth range of 0.1–15 nm. These filters were used to emulate a Mux/DeMux by allowing the testing of any number of comb lines individually or collectively. The combs were setup using a Vescent D2-500 low-noise laser controller, a Vescent SLICE QTC TEC, and a Keithley 2602B power supply. The pulses were amplified by a 26 dB praseodymium-doped fiber amplifier (PDFA) to increase the power per line thus compensating for the 7 dB insertion loss (IL) of TBF and the lack of TIA at the receiver.

At the transmitter, we use a Keysight M8199B arbitrary waveform generator (AWG) with two 256 GSa/s DAC channels. First, a random sequence of QAM symbols is generated. The signal is then shaped using a root raised cosine (RRC) filter at two samples per symbol (sps) and resampled to match the rate of the AWG. Multi-line experiments use an RRC roll-off factor of 0.02 to respect the separation between the adjacent comb lines. Next, we pre-compensate the frequency response of the AWG and the RF cables using a digital pre-emphasis filter. The signal is then clipped to limit its peak-to-average power ratio to 9 dB at 56 Gbaud.

We use a 100 GHz GSG-GSG RF probe to drive a thin-film lithium niobate (TFLN) IQ modulator (IQM). The IQM is a single polarization Mach-Zehnder modulator with a fiber-to-fiber insertion loss of 9 dB. The TFLN IQM has a $V_\pi$ of 1.7 V and 3 dB bandwidth of ~30 GHz, however its small roll-off allows for operation over 100 GHz[44]. The modulator is biased manually by tuning the individual phase shifters on the I and Q branches of the IQM. The bias is optimized by measuring the resulting BER and constellation during the experiment. This method was used due to a lack of automatic bias controller and could lead to some bias errors. Additionally, the extinction ration of the modulator was relatively low, leading to some additional leaking of the carrier. The output is then amplified by a PDFA to compensate for not using transimpedance amplifiers at the receiver. We then employ a dual-polarization (DP) emulator by using a polarization controller (PC) and a polarization beam splitter (PBS) which split the power into two orthogonal polarizations. One polarization is then delayed by 9.2 ns using a variable optical delay line (VODL) to decorrelate the symbols on each polarization. Finally, the signal is transmitted over 10 km of SMF.

At the receiver (Rx), we use an O-band DP optical hybrid whose output is connected to four 70 GHz balanced photodiodes (BPDs) and finally to the RTO. The LO consists of either a tunable ECL, an array of DFBs, or a second QD-MLL. The ECL and DFBs both had an output power of 15 dBm, and a linewidth of ~500 kHz and 1 MHz, respectively. The second QD-MLL was connected to an EXFO XTM-50 narrow TBF and an 18 dB PDFA to select a single comb line. The ROP at the BPD was kept constant at 5 dBm. The DSP at the receiver is performed offline and consists of first deskewing the received signals for each polarization, correcting the frequency offset, and RRC filtering the signal to 2 sps. The equalizer used is a time-domain 4 × 4 real-valued MIMO equalizer (Butterfly form) that uses the LMS algorithm and mean square error between training symbols and the received symbols after the channel as a criterion. The MIMO is further interleaved with a 2nd order phase locked loop that uses the phase error between the training and received symbols and offers fine tracking of the carrier phase and the frequency offset[62]. This equalizer structure enables compensating CD, FO, and phase noise, but with very limited range as seen from our data for FO (0.035% of the symbol rate). Finally, the symbols are converted into a bit sequence to calculate the BER.

### Detailed IMDD experimental setup

Supplementary Figure S18 shows the IMDD optical fiber transmission system (OFTS) experimental setup. The dual-polarization emulator, the optical 2 × 8 hybrid, and the second comb used as LO are removed. The transmitter (Tx) source consisted of a QD-MLL connected to an O-band Santec OTF-350 optical tuneable bandpass filter (TBF) with a bandwidth range of 0.1–15 nm and allowing the testing of any number of comb lines individually or collectively. The bandwidth of the TBF after the comb source is set to around 180 GHz. This allows three comb lines to be bulk modulated at once. This prevents saturating the PDFAs while still allowing for a neighbor on each side of the channel under test. The pulses were then amplified by a 26 dB praseodymium-doped fiber amplifier (PDFA) to increase the power per line thus compensating for the 7 dB insertion loss (IL) of the Santec TBF. At the transmitter, we use a Keysight M8199B arbitrary waveform generator (AWG) with one 256 GSa/s digital-to-analog (DAC) channel. First, $2^{19}$ PAM symbols are generated. Multi-line experiments use a RC roll-off factor of 0.02 to respect the separation between the adjacent comb lines. Next, we pre-compensate the frequency response of the AWG and the RF cables using a digital pre-emphasis filter. We use a 100 GHz GSG-GSG RF probe to drive a thin-film lithium niobate (TFLN) Mach-Zehnder modulator (MZM). The TFLN MZM has a $V_\pi$ of 1.7 V and 3 dB bandwidth of ~30 GHz, however its small roll-off allows for operation over 100 GHz. The signal is then transmitted over 10 km of SSMF.

At the receiver (Rx), we use an EXFO O-band XTM-50 narrow TBF to select a single comb line. The bandwidth of the second TBF is set to ~58 GHz. This allows this TBF to act as a perfectly centered demux that mitigates the effects of neighboring channels at the receiver. The single channel is then amplified by a second PDFA whose output is connected to a 70 GHz photodiode (PD) and finally to the RTO. The received optical power is monitored using the second PDFA which is also used to compensate for not using transimpedance amplifiers at the receiver and for the 5 dB IL of the EXFO TBF. The ROP of the single channel was kept constant at 2 dBm. At the receiver, the capture from the photodetector is digitized by an ADC such that the DSP can be performed offline. The DSP at the receiver consists of first resampling the signal to 2 sps and employing a feed-forward equalizer (FFE). Finally, the symbols are converted into a bit sequence to calculate the BER.

### Comb characterization measurement setup

The comb was setup using a Vescent SLICE QTC TEC and a Keithley 2602 A power supply to test the laser properties at various comb states. To test the spacing difference between the different comb lines for a single state, the comb laser spacing was measured using a 50 GHz Rhode & Schwarz FSU50 ESA and a 39 GHz external Marki MM12567L mixer. The comb source was input into a TBF such that only 2 consecutive lines were output. The filter was tuned to sweep across all 28 lines. For each pair, 10 ESA traces were measured and the comb laser spacing, and linewidth were determined. The spacings were placed in 40 kHz bins and plotted as a histogram. The experiment was repeated using a Vescent D2-500 low noise controller for more accurate measurements.

### Detailed simulation setup

We have carried out transmission system simulations using Optiwave's Optisystem to investigate the nature and onset of the penalty that can be ascribed to nonlinear effects in the analyzed comb-based architectures. We have focused on Kerr nonlinearities and disregarded the effect of stimulated Brillouin scattering since this impairment is normally reduced using well-known optical frequency dithering techniques[63]. Coherent and IMDD OFTS were compared for 10 km reach and 56 GBaud transmission in O-band and C-band. The zero-dispersion wavelength was positioned at the center of the channel plan, corresponding to 1299 nm. This is the worst scenario to induce fiber nonlinear behavior. The dispersion slope used corresponds to an ITU-T G.652 fiber, which in this case was SMF-28. The value was

0.09 ps/nm²·km. The PMF coefficient used was 0.05 ps/sqrt(km) which is commonly found in fibers.

The BER performance of the central wavelength of the comb was analyzed and the resulting SNR penalty was calculated. We assumed the central wavelength was centered in the zero-dispersion wavelength of a conventional SMF (dispersion slope of 0.09 ps/nm²·km, $\alpha = 0.35$ dB/km and $n_2 = 2.4 \times 10^{-20}$ m²/W). In all cases, the reach was set to 10 km, and the symbol rate to 56 Gbaud. The laser/comb linewidth was set to 200 kHz to match the initial comb characterization. We compared a coherent OFTS with 2 separate comb sources for carrier and LO to an IMDD OFTS. For the coherent case, the modulation was set to DP-16QAM and an OSNR of 19 dB was selected. A conventional DSP-aided coherent receiver was employed where the input power into the balanced photodetector was always maintained at +6 dBm (-10 dBm at the input of the 90° hybrid) to keep constant the noise-induced receiver penalty. We suppressed linear XT by setting a Mux and DeMux with (brick-wall) filter bandwidths (BW) of 60 GHz. This was validated through a performance comparison of the multi-line system operating at low input power with the linear ($n_2 = 0$) single-channel case. PAM4 was the modulation of choice for the IMDD investigation. For comparison purposes, the OSNR of the system was adjusted to agree with the BER performance of the coherent system under linear operating conditions. The input power into the photodetector was also maintained at +6 dBm. We suppressed linear XT by setting a Mux and DeMux with (brick-wall) filter BW of 56 GHz.

## Data availability
The data of this study are available from the corresponding author upon request.

## Code availability
The codes used in this study are available from the corresponding author upon request.

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

## Acknowledgements

The McGill University contributions were supported by the Natural Sciences and Engineering Research Council of Canada (NSERC), the Canadian Foundation for Innovation (CFI), the Fonds de recherche du Québec Nature et Technologies (FRQNT), and McGill University. The UCSB contributions were funded by the DARPA PIPES Program (HR0011-19-C-0083). The authors would also like to thank Hyperlight, Anritsu, TestForce, EXFO, and Santec.

## Author contributions

S.B. and E.B. designed and built the experimental setups with assistance from M.D., C.S.A., W.L., Z.W., B.K. and F.P. S.B. performed the transmission experiments and analyzed the data. The simulations were performed by R.G.C. and Y.H. with assistance from E.B. and S.B. M.D. designed, fabricated, and performed initial characterization of the comb. S.B. drafted the manuscript with support from all co-authors. J.B. and D.V.P. supervised the work.

## Competing interests

The authors declare no competing interests.
