## [Peer Review File · Nature Communications]

12.1 Terabit/Second Data Center Interconnects Using O-band Coherent Transmission with QD-MLL Frequency CombsREVIEWER COMMENTS

Reviewer #1 (Remarks to the Author):

This paper presents the use of QD-MLL comb source as a suitable source for coherent transmission in the O-band and describes an experiment showing coherent transmission using this laser, achieving 12 Tbit/s over 10km. The authors claim this is a 3-fold increase in throughput over the use of quantum-dash MLL. The introduction to the paper is a rather general explanation of coherent transmission vs IMDD and the use of modelocked lasers (MLL) for coherent transmission. Although it may be of interest to those outside the field, it brings little information to those working in the field. The table in Fig 1 is a nice summary but it is unnecessarily narrow - why limit this to experiments involving at least one mode-locked laser? Why exclude work which has demonstrated much greater throughputs and distances in the O-band - eg from KDDI Labs (Elson et al)? What is the advantage of comb-to-comb configurations? irrespective what type of comb sources are used? The authors argue energy and integration - but there is no direct comparison of different types of sources so whilst the experiment is very impressive - it is not clear that this type of comb source when used in the transmitter or local oscillator or both is ideal. The experiment in section 4.3 gives very specific results for the configuration used but does not generalise them.

The authors do not really study its stability, phase noise nor fundamental limits that could be achieved with such sources vs alternative sources. The section on nonlinearities does not explain what the likely nonlinear limits are over 10km?

Finally, section 5 on the discussion and the 'roadmap' is a rather grand claim. The analysis is interesting but far from exhaustive comparison of different comb and other sources (which would have been of tremendous interest if more complete).

Overall, this is a very elegant & impressive experiment achieving a very good result for this specific laser and the paper claims, title and abstract do not reflect the specificity of the achievement. Very good work, deserves publication but please change title which is too general - specify which type of comb is being used and please change claims about a 'roadmap' wherever it appears - there is no roadmap, just some preliminary estimates and indications. Please reference much higher rates & distances achieved through the use of ECLs by KDDI - Elson et al (OFC 2023 is cited, but only in reference to the nonlinearities - - but not more generally, not cited ECOC2023 nor multiple of their journal papers).

Reviewer #2 (Remarks to the Author):

Review comments:

This paper proposes and reports on an O-band multichannel coherent and IMDD transmission targeting data center scenarios up to 10km coverage. The key enabling technology is O-band QD-MLL based comb source placed on both the transmitter side and the receiver side working as source and LO respectively. Up to 28 channels of DP-16QAM and up to 26 channels of DP-32QAM, both up to 56 Gbaud, were transmitted with the proposed scheme. Very high symbol rate can also be achieved at single channel with the QD-MLL source with only single carrier filtered out. The overall data rate of the single fiber system is up to 12.1 Tbps. This work is also supported with many analytical and numerical simulations. While the quality of the work and the validity of the experiment are clearly good, I do have several critical questions regarding the motivation of the scheme itself, the main novelty of the work, and technical stringency of the demonstration. Besides, there are a few technical questions for the authors to address.

Main concerns:

1. The authors motivated the scheme with the arguments of challenges such as power, complexity and cost when further scaling up the data rates beyond Tbps. Nowadays the DCI (intra- and inter-

DC) transmissions are dominated by optical pluggable transceivers. One of the advantages of the pluggables is the ease of replacement upon failures. The proposed scheme needs integration of 20+ IQ modulators together with the MUX and DEMUX either monolithically with the QD-MLL on a single PIC, or heterogeneously combine them on chiplets inside a module. Either way, upon failure of a single modulator, the cost and complexity of replacement will be higher than the pluggables. The integration of amplifiers further blurred the advantages of power saving compared with separate laser sources. I recommend the authors carefully justify such a scheme from the application point of view.

2. Regarding novelty, it's unclear what special techniques the authors used in this work made the overall transmission rate higher than the C-band demonstrations. Is there any breakthrough/novel design of the QD-MLL itself compared with previous C-band designs? If it is only the optimization of the number of comb lines vs line power, then the novelty is limited. On the other hand, we do see smaller nonlinear penalties in the C-band by looking at Fig. 6b. Please clarify the key enabling techniques that empowered this record.

3. Regarding the technical stringency of this work, the biggest concern is the "sliding window and bulk modulation" scheme. In this way, the central carrier and the aggressors (both in coherent and in IMDD) are all modulated with the same signals and transmitted simultaneously. The impact of linear/nonlinear crosstalk can easily become underestimated or even ignored. It has been a common practice to at least decorrelate even and odd channels for DWDM transmissions. The authors need to carefully justify this scheme with quantifications.

Other technical comments:

1. Three PDFAs are used in the transmission as booster, as pre-amplifier and for LO amplification. And this is only for 5 channels. Please justify quantitatively the benefit in terms of power consumption of such scheme compared with single laser DWDM configurations. Now it's likely for single-wavelength O-band coherent transceivers to operate without optical amplifications considering the limited fiber loss.

2. I don't quite get how the adaptive equalizer can remove residual FO up to 20 MHz. Typical adaptive equalizers (e.g., CMA) only focus on amplitude and don't touch on the phase of the symbols. Please specify what exactly the configuration of the 4x4 MIMO equalizer. Does it include also a phase noise compensation process?

3. The 10% reduction in DSP power consumption with O-band compared with C-band is not very clearly justified. For 10km distance at 56 Gbaud, typical adaptive equalizers with a bit longer than 10-symbol memory length can also compensate CD quite well in the C-band.

4. How was the IQ modulator biased? Figure 4c shows central peaks for modulated carriers. It seems like the carriers were not suppressed like conventional QAM modulation.

5. For the IMDD transmission, how much was the bandwidth of the TBF?

6. Again for IMDD, was the 2dBm ROP per channel or per 3 channels? And how much was it for coherent ROP? In fact, considering PDFAs are used for the signal, one should really show OSNR than ROP values as the signals are additive noise limited rather than receiver noise limited.

Other comments:

1. In the Abstract, when claiming 'highest capacity C-band systems reported to date', please add condition that it is compared with QD-MLL based C-band systems. Record transmission rate of C-band systems in general is much higher.

2. Some typos are spotted in the introduction, 'Gpbs'  'Gbps'.

3. The linewidth color bar shown in Fig. 3a may be better shown with log scale, as it is more interesting to focus on low-linewidth region below 500 kHz, which is all blue in the current figure.

4. In first paragraph of Sec. 4.1, some figure numbers are missing (Fig. b, Fig. c).

5. On page 21, The sentence 'By increasing the ROP of the DFB from 2 dBm to 2.7 dBm, ...' is confusing.

6. On page 22, there is a statement of "The signal bandwidth cannot be larger than the spacing between comb lines due to the penalty of inter symbol interference (ISI)". I am not sure if ISI is the correct term here.

7. In the first paragraph of Sec. 5, 100 bps/lambda should be 100 Gbps/lambda.

Reviewer #3 (Remarks to the Author):

The submitted manuscript discusses the use of frequency combs for ultra-high speed intra-datacenter communications.

The work is generally very relevant wrt next generation intra DCN.

The manuscript is very well written and concise, however, there are a number of aspects the authors could improve.

- it would be good to mention the fiber parameters. Especially relevant are exact position of dispersion zero, dispersion slope and PMD value.
- in section 2.1 IM/DD it would be good to already mention FWM as an important impairment hindering tighter channel grids in IM/DD high speed transmission in the O-Band. Also some discussions could be added on recently proposed ways of improving FWM penalties (e.g. orthogonal polarization launch of neighboring channels)
- it would be interesting to add a short discussion about system parameter differences between single channel lasers and QD-MLLs such as total power consumption and heat dissipation.
- in section 3 the power conversion efficiency is mentioned to be high. It would be good to quantify (and compare to "normal" lasers)
- in Fig. 4 some constellations show non-round (i.e. nonlinear) shape. Can you give an explanation?
- in Fig. 6 a comparison of NL penalty is shown. In recent research of 1.6TbE+ systems, orthogonal (polarization) launch is discussed to significantly reduce FWM penalties. Could you please add a discussion and also explain whether this could also be realized in your comb system setup. Also the effect of PMD will have a significant impact on FWM penalties as it leads to depolarization along the transmission line. This may be difficult to evaluate experimentally, though. Did you see any temporal variations of the FWM penalty?
- it would be interesting to mention also which exact DSP has been used. Please give some details about filter lengths, etc. In Fig. 9 no equalizer (FFE/DFE) is shown. Is that correct? Or is it included in the MIMO eq? Later on it is referred to "a conventional DSP" . Please specify in more details.
- In Fig. 10 an FFE is shown. In the text offline DSP is mentioned. Please give details.
- In the supplement a working temperature of 42°C is mentioned. Apart from wavelength drift, what effect has a change in temperature on optical power?
- Could you add an axis for temperature variation in S7 (e.g. second x-axis)?

Responses to Reviewer comments

Reviewer 1:

Reviewer #1 (Remarks to the Author):

1.1 This paper presents the use of QD-MLL comb source as a suitable source for coherent transmission in the O-band and describes an experiment showing coherent transmission using this laser, achieving 12 Tbit/s over 10km. The authors claim this is a 3-fold increase in throughput over the use of quantum-dash MLL. The introduction to the paper is a rather general explanation of coherent transmission vs IMDD and the use of modelocked lasers (MLL) for coherent transmission. Although it may be of interest to those outside the field, it brings little information to those working in the field.

We thank the reviewer for the summary of our work and the insightful comment. We originally focused our introduction on the comparison between coherent and IMDD due to the specific use case being investigated (O-band, 10km DCI). This use case has been traditionally dominated by IMDD. We therefore saw it necessary to introduce and address the important distinctions between IMDD and coherent before establishing our suggested comb-based coherent architecture. As is somewhat common for Nature Communications articles, we felt it appropriate to situate the work early in the discussion to those outside the field.

Additionally, we expanded section 3 to highlight some of the advantages of MLL as compared to other comb sources as expressed also from reviewer comments 1.3. This will add more context to the experimental work and provide a more insightful background for the work presented.

1.2 The table in Fig 1 is a nice summary but it is unnecessarily narrow - why limit this to experiments involving at least one mode-locked laser? Why exclude work which has demonstrated much greater throughputs and distances in the O-band - eg from KDDI Labs (Elson et al)?

We thank the reviewer for the question and suggestion. In this first manuscript, we limited the scope of the table to coherent transmission demonstrations over standard single mode fiber (SSMF) using quantum-well, quantum-dash, and quantum-dot mode-locked lasers as a means of comparing our work to published experimental demonstrations employing similar laser source configurations and propagating fiber. However, we welcome the reviewer suggestion to increase the focus of the comparison and included more references to Fig 1. The result is a more compressive and therefore more useful comparison of reported comb laser technologies while focusing the discussion to use cases that employ i) a single comb laser for the carrier, and ii) propagation over SSMF.

In this context, we chose not to include the Elson et. al., publications in Figure 1 because they do not overlap with the use case of our submission. We do, however, acknowledge the importance of their work and thus added a brief description in the introduction about the high throughput over long distances achievable using WDM systems with multicore fibers [1,2] and higher bandwidths [3,4]. We welcome additional references that should be cited.

[1] D. J. Elson et al., "115.2-THz aggregate bandwidth O-band coherent DWDM transmission over high-density 250- μm coating multicore fibre," 49th European Conference on Optical Communications (ECOC 2023), Hybrid Conference, Glasgow, UK, 2023, pp. 1662-1665, doi: 10.1049/icp.2023.2661.

[2] D. J. Elson, Y. Wakayama, S. Beppu, D. Soma and N. Yoshikane, "Transmission of 400GBASE-LR8 Over 15 km Deployed Step-Index 4-Core fiber for Data Centre Interconnects," 2022 Optical Fiber Communications Conference and Exhibition (OFC), San Diego, CA, USA, 2022, pp. 1-3.

[3] Y. Wakayama et al., "400GBASE-LR4 and 400GBASE-LR8 Transmission Reach Maximization Using Bismuth-Doped Fiber Amplifiers," in *Journal of Lightwave Technology*, vol. 41, no. 12, pp. 3908-3915, 15 June 2023, doi: 10.1109/JLT.2023.3269536.

[4] D. Soma et al., "25-THz O+S+C+L+U-band digital coherent DWDM transmission using a deployed fibre-optic cable," 49th European Conference on Optical Communications (ECOC 2023), Hybrid Conference, Glasgow, UK, 2023, pp. 1658-1661, doi: 10.1049/icp.2023.2660.

1.3 What is the advantage of comb-to-comb configurations? irrespective what type of comb sources are used? The authors argue energy and integration - but there is no direct comparison of different types of sources so whilst the experiment is very impressive - it is not clear that this type of comb source when used in the transmitter or local oscillator or both is ideal. The experiment in section 4.3 gives very specific results for the configuration used but does not generalise them.

We thank the reviewer for the question. The advantage of using a comb source for coherent systems is explained in section 2.2 and 3 of the paper. In brief, by using a single QD-MLL comb source as a transmitter, we can use a single TEC to stabilize all the transmitter comb laser lines/ λ 's. Additionally, receiver DSP blocks found in a traditional coherent receiver can either be removed (e.g., no chromatic dispersion compensation because of O-band operation) or simplified (e.g., carrier phase recover). As was pointed out by Eric Maniloff from Ciena at OFC 2024 [1], reductions and/or simplifications create opportunities to re-optimize DSP architectures [Ciena OFC 2024]. Additionally, including a second comb lasers enables deployment of full bi-directional coherent optical fiber transmission system wherein each comb laser is used as i) the upstream carrier and ii) the downstream local oscillator. In sum, as noted above, this system architecture therefore requires only two TECs, one each for the downstream an upstream carrier/LO. An additional power analysis was added to section 5 to quantitatively compare single- λ sources to QD-MLL comb sources.

We thank the reviewer for the comment. We agree with the need to generalise the comb-to-comb results to showcase the benefits of the system irrespective of the comb used. Accordingly, we included a generalisation of our comb-to-comb results whilst promoting the

QD-MLLs as the most viable option. To do so and as mentioned in response 1.1, we expanded section 3 to also compare other comb sources such as Kerr solitons microcombs [2]. As a means of quantifying performance, we utilized our presented figure of merit, namely the Rate of Side Mode Suppression (RSMS), as means for comparison.

[1] E. Maniloff, "Design Tradeoffs for Coherent Pluggable Optics At 800G and Beyond," OFC, 2024.

[2] Geng, Y., Zhou, H., Han, X. et al. Coherent optical communications using coherence-cloned Kerr soliton microcombs. Nat Commun 13, 1070 (2022). <https://doi.org/10.1038/s41467-022-28712-y>

1.4 The authors do not really study its stability, phase noise nor fundamental limits that could be achieved with such sources vs alternative sources.

We thank the reviewer for pointing out this gap in our explanations. We agree with the need to provide more details on the stability and phase noise of the laser used in our experiments. To that end, we referenced the previous work of our collaborators [1] including adding additional MLL comb source performance details in section 3 of the main text. Regarding phase noise comparisons, we avoided this topic because it would be challenging quantitatively to perform this comparison whilst maintaining manuscript focus, which is on coherent optical fiber transmission system performance using QD-MLLs.

[1] Zibar, Darko; Razumov, Aleksandr; Heebøll, Holger; Dumont, Mario; Terra, Osama; Dong, Bozhang; et al. (2023). Subspace tracking for phase noise source separation in frequency combs. Optica Open. Preprint. <https://doi.org/10.1364/opticaopen.24155511.v1>

1.5 The section on nonlinearities does not explain what the likely nonlinear limits are over 10km?

We thank the reviewer for this question, and we modified section 4.2 in the main text with the following text.

For the simulations in this paper, we only considered the nonlinear (NL) effects at 10km as this was the use case of our study. We would expect the NL effects to diminish for distances greater than 10 km because the light power traveling in the fiber reduces over distance. This effect is magnified by the higher attenuation in the O-band as compared to the C-band as can be seen in figure 6b. The limit for our use case of coherent transmission in the O-band over 10 km is ~ 6 dBm of total input power into the fiber. The resulting SNR penalty from the NL effects induced at this power cause an increase in the BER over the targeted FEC overheads. This would result in a throughput reduction from 10.18 Tbps to 9.71 Tbps and from 12.14 Tbps to 11.65 Tbps for the DP-16QAM and DP-32QAM results, respectively.

1.6 Finally, section 5 on the discussion and the 'roadmap' is a rather grand claim. The analysis is interesting but far from exhaustive comparison of different comb and other sources (which would have been of tremendous interest if more complete).

We thank the reviewer for the insightful comment and agree that a comparison with other comb sources would be very valuable. In our initial analysis, we focused on comparing different QD-MLLs comb sources to make a fair comparison based on the experimental results obtained (using only QD-MLL comb sources) and the assumptions established in the paper. For Table 2, the constants were: 1) a fixed optical bandwidth of 1.2 THz and 2) the same achievable throughput for all examples. In response to this suggestion, we modified the text to expand this comparison table to comb based systems in general as the assumptions can be valid when using other comb sources.

For Table 3, the boundary comparison points were: 1) a fixed optical bandwidth of 1.2 THz, 2) DP-32QAM operation, 3) 6 quantum-dot layer structure, 4) second harmonic MLL operation, and 5) operating temperatures between 20°C and 60°C. Therefore, we did not add other sources directly to the table. However, we did expand the text to suggest some of the limitations of the QD-MLL variations that can be overcome using different comb source technologies such as the large spacing requirements over 100 GHz [1].

Additionally, we added a quantitative analysis of the power consumption of such a system as compared to single- λ laser sources as requested by reviewer comment 2.4.

[1] Marin-Palomo, P., Kemal, J., Karpov, M. et al. Microresonator-based solitons for massively parallel coherent optical communications. *Nature* 546, 274–279 (2017). <https://doi.org/10.1038/nature22387>

1.7 Overall, this is a very elegant & impressive experiment achieving a very good result for this specific laser and the paper claims, title and abstract do not reflect the specificity of the achievement. Very good work, deserves publication but please change title which is too general - specify which type of comb is being used and please change claims about a 'roadmap' wherever it appears - there is no roadmap, just some preliminary estimates and indications.

We thank the reviewer for the kind words and suggestion. We changed the title to be more specific to our comb source and we reduced the roadmap claims in the text.

The new title is: "12.1 Terabit/Second Data Center Interconnects Using O-band Coherent Transmission with QD-MLL Frequency Combs"

1.8 Please reference much higher rates & distances achieved through the use of ECLs by KDDI - Elson et al (OFC 2023 is cited, but only in reference to the nonlinearities - - but not more generally, not cited ECOC2023 nor multiple of their journal papers).

We thank the reviewer for the suggestion. Per comment 1.2, we cited the following KDDI works [1-4] in section 1 of the main text.

- [1] D. J. Elson et al., "115.2-THz aggregate bandwidth O-band coherent DWDM transmission over high-density 250- μm coating multicore fibre," 49th European Conference on Optical Communications (ECOC 2023), Hybrid Conference, Glasgow, UK, 2023, pp. 1662-1665, doi: 10.1049/icp.2023.2661.
- [2] D. J. Elson, Y. Wakayama, S. Beppu, D. Soma and N. Yoshikane, "Transmission of 400GBASE-LR8 Over 15 km Deployed Step-Index 4-Core fiber for Data Centre Interconnects," 2022 Optical Fiber Communications Conference and Exhibition (OFC), San Diego, CA, USA, 2022, pp. 1-3.
- [3] Y. Wakayama et al., "400GBASE-LR4 and 400GBASE-LR8 Transmission Reach Maximization Using Bismuth-Doped Fiber Amplifiers," in *Journal of Lightwave Technology*, vol. 41, no. 12, pp. 3908-3915, 15 June 2023, doi: 10.1109/JLT.2023.3269536.
- [4] D. Soma et al., "25-THz O+S+C+L+U-band digital coherent DWDM transmission using a deployed fibre-optic cable," 49th European Conference on Optical Communications (ECOC 2023), Hybrid Conference, Glasgow, UK, 2023, pp. 1658-1661, doi: 10.1049/icp.2023.2660.

Reviewer 2:

Reviewer #2 (Remarks to the Author):

Reviewer comments:

This paper proposes and reports on an O-band multichannel coherent and IMDD transmission targeting data center scenarios up to 10km coverage. The key enabling technology is O-band QD-MLL based comb source placed on both the transmitter side and the receiver side working as source and LO respectively. Up to 28 channels of DP-16QAM and up to 26 channels of DP-32QAM, both up to 56 Gbaud, were transmitted with the proposed scheme. Very high symbol rate can also be achieved at single channel with the QD-MLL source with only single carrier filtered out. The overall data rate of the single fiber system is up to 12.1 Tbps. This work is also supported with many analytical and numerical simulations. While the quality of the work and the validity of the experiment are clearly good, I do have several critical questions regarding the motivation of the scheme itself, the main novelty of the work, and technical stringency of the demonstration. Besides, there are a few technical questions for the authors to address.

We thank the reviewer for the insightful summary of our work, kind words, and feedback.

Main concerns:

2.1. The authors motivated the scheme with the arguments of challenges such as power, complexity and cost when further scaling up the data rates beyond Tbps. Nowadays the DCI (intra- and inter-DC) transmissions are dominated by optical pluggable transceivers. One of the advantages of the pluggables is the ease of replacement upon failures. The proposed scheme needs integration of 20+ IQ modulators together with the MUX and DEMUX either monolithically with the QD-MLL on a single PIC, or heterogeneously combine them on chiplets inside a module. Either way, upon failure of a single modulator, the cost and complexity of replacement will be higher than the pluggables. The integration of amplifiers

further blurred the advantages of power saving compared with separate laser sources. I recommend the authors carefully justify such a scheme from the application point of view.

We thank the reviewer for the insightful comment. We agree that the complete integration of a QD-MLL represents a potential single point of failure in our presentation of a densely integrated Photonic Integrated Circuit (PIC). We believe that over time PIC technologies will become (indeed must become) increasingly more reliable on route to widespread adoption. In this context, we note that Table 2 illustrates that our current schematic is compatible with current 400G systems which are commercially available now. Additionally, the proposed solution is ~ 3 to 4 generations ahead of current Ethernet node requirements meaning that its implementation should coincide with more mature and reliable 400G components. This in turn would increase the production yield and shelf-life of such a scheme. We also agree that the current pluggable market will not shift immediately to our proposed scheme. Therefore, and in response to this comment/suggestion, we modified the main text to explain this limitation and added the following section to the supplemental to support the implementation of QD-MLLs in current DCI coherent systems using pluggables.

Main:

This concept is limited by the potential high replacement cost of a single failure in such a densely integrated PIC. This risk can be mitigated by using more mature and reliable standard components as further detailed in section 5. This would increase the production yield and shelf-life of such a scheme. On the other hand, this risk can be removed by operating with more commercially available components such as optical pluggable transceivers and receivers which already dominate the DCI space. This concept is described in supplemental section 3. This would allow for individual pluggables to be replaced as needed in case of failures.

Supp:

S3: Comb-based DCIs using pluggables

Figure S2a illustrates the concept of a single QD-MLL enabling a coherent comb-to-comb optical transmission system using n -pluggables in a current data center rack configuration. This configuration is composed of a top rack with n -pluggables and a lower rack with the comb laser source. Fig. S2c shows the comb laser and demultiplexer which corresponds to a single rack of the concept shown in Fig. S2a. The system would only require one TEC to stabilize the laser source for all pluggables. The laser source output would be de-multiplexed and transmitted separately to all n -pluggables to be used as both carrier (outbound) and LO (inbound). This is shown in Fig S2a by the fibers connecting the bottom rack to the top rack. This would require an upgrade to the current pluggable form factors as the need for an external laser source would require a third optical interface, as compared with current QSFP standards. This form factor evolution is not uncommon as technologies improve as was seen with the introduction of QSFP replacing SFP form factors for higher Ethernet standards. As seen in Figure S1b, this 3-port pluggable would amplify the external laser source and split it to be used as a carrier and LO. A Tx PIC would modulate the signals and output them to be

multiplexed by current WDM Tx filter racks to be propagated in a single fiber. The inbound signals would be de-multiplexed and transmitted to each receiver pluggable using the already existing WDM Rx filter racks. This concept would require careful planning of the layout of the different fibers in order to ensure a small form factor for DCI operations.

Figure S2: **Concept of a comb-to-comb coherent scheme consists of n -pluggables driven by one QD-MLL.** The datacenter rack a is composed of n external laser pluggables b driven by a single QD-MLL c.

We also agree that introducing of SOAs may reduce some of the power savings arrived at when implanting the QD-MLL. However, we note that every line does not need its own amplifier. We recently demonstrated that a single quantum-dot semiconductor optical amplifier (QD-SOA) can be used to operate high speed WDM coherent systems [1]. In this paper, a single SOA was used for 4 lines. Thus, our scheme does not necessarily require 20+ SOAs. A more detailed SOA power analysis is presented in reviewer response 2.4 and was added to the main text in section 5.

[1] C. St-Arnault, S. Bernal, et. al, "Net 1.6 Tbps (4x400Gbps/ λ) O-Band IM/DD transmission over 2 km using uncooled DFB lasers on the LAN-WDM grid and sub-1V drive TFLN modulators" Optical Fiber Communication Conference (OFC), post-deadline paper, Th4C.6 (2024)

2.2. Regarding novelty, it's unclear what special techniques the authors used in this work made the overall transmission rate higher than the C-band demonstrations. Is there any breakthrough/novel design of the QD-MLL itself compared with previous C-band designs? If it is only the optimization of the number of comb lines vs line power, then the novelty is limited. On the other hand, we do see smaller nonlinear penalties in the C-band by looking at Fig. 6b. Please clarify the key enabling techniques that empowered this record.

We thank the reviewer for the insightful question and comment. The novelty in this work was enabled by integration of an O-band, 60 GHz spacing, quantum-dot (as opposed to previously demonstrated quantum-dash or quantum-well) MLL comb source as a carrier and LO for a novel use case (O-band coherent for 10 km in DCI). The laser was fabricated with a larger comb line spacing which allowed the use of higher symbol rates. Additionally, the low phase noise and stability of the laser permitted the use of higher order modulation formats which allowed us to achieve higher throughputs. We modified section 1 to further clarify this point.

2.3. Regarding the technical stringency of this work, the biggest concern is the "sliding window and bulk modulation" scheme. In this way, the central carrier and the aggressors (both in coherent and in IMDD) are all modulated with the same signals and transmitted simultaneously. The impact of linear/nonlinear crosstalk can easily become underestimated or even ignored. It has been a common practice to at least decorrelate even and odd channels for DWDM transmissions. The authors need to carefully justify this scheme with quantifications.

We thank the reviewer for the insightful comment. We agree that the use of bulk modulation is not the ideal configuration for this type of DWDM experiment. However, the biggest penalty of bulk modulating is incurred in the distribution of the RF signal over multiple λ 's. This can be observed by the order of magnitude difference in BER at 56 GBaud between the single- λ (Fig. 8a) and bulk-modulated (Fig. 4a) experimental results shown in section 4 of the main text.

The use of bulk modulation was required due to the limitations of commercially available O-band components. For example, most DWDM experiments that utilize even/odd schemes operate in the C-band and use an optical waveshaper as a programmable optical filter to curate λ 's. Unfortunately, there was no O-band waveshaper available on the market at the time of this experiment. Note that Coherent has since released an O-band version of their waveshaper instrument [1]. Additionally, custom O-band multiplexers and demultiplexers were not available to us at the time of the experiment to test each line separately.

In response to this comment/suggestion, we added the following section in the supplemental to justify our use of bulk-modulation:

Supp:

We performed simulations to calculate the difference between bulk and non-bulk modulated results for both IMDD and coherent. For IMDD, we found that the linear crosstalk was less than 0.1 dB due to the demux acting as a perfect bandpass filter. Figure S13 shows the calculated variations in SNR penalty as the central wavelength of the demux is shifted for a 56 Gbaud PAM 4 signal after 10 km. Our results show that the linear cross talk difference between bulk and non-bulk modulation in the IMDD case is only present when the neighbouring signals are also detected by the receiver. These results were experimentally validated as can be seen in Fig S14. In our IMDD experiment, we utilised a tunable bandpass filter (TBF) to act as a perfectly centered demux that mitigates the effects of neighboring channels at the receiver. Our results show that the BER improves when the neighboring signal present at the detector has the same bits as the device under test.

Figure S13: Calculated BER penalty as a function of a demux frequency shift for either bulk or non-bulk modulated signals. A comb with 5 lines was used as transmitter with a line spacing of 58 GHz.

Figure S14: Measured BER penalty as a function of a temperature-induced frequency shift for bulk modulated signals. A comb with 5 lines was used as transmitter with a line spacing of 58 GHz.

Our results also show that the linear cross talk is stronger for the bulk modulated signal in the coherent case. Figure S15 shows the calculated SNR penalty as a function of signal roll-off. As

the roll-off increases, the SNR penalty increases due to the linear cross talk between neighbouring channels. For our experiment, we used a roll-off of 0.03 to minimize this effect which we calculated to be less than 0.05 dB. We would expect a slightly better performance if the experiment was done with a non-bulk modulated setup. Additionally, the presence of the LO acting as a bandpass filter enables the removal of any noise from the neighbouring channels as can be seen in Fig S16. Our results show that the BER remains constant independently of the number of lines injected into the coherent receiver. This shows the tolerance of coherent detection to linear crosstalk. The variations in the BER are mainly due to measurement variations for that experiment.

Figure S15: **Calculated BER penalty as a function of a Tx roll-off for either bulk or non-bulk modulated signals.** A comb with 5 lines was used as transmitter with a line spacing of 58 GHz.

Figure S16: **Measured BER penalty as a function of a demux bandwidth increase for bulk modulated signals.** A comb with 5 lines was used as transmitter with a line spacing of 58 GHz.

[1] WAVESHAPER 500B/1000B/4000B. (2024). Accessed: May 2 2024. [Online]. Available: <https://www.coherent.com/resources/datasheet/networking/waveshaper-500b-1000b-4000b-ds.pdf>

Other technical comments:

2.4. Three PDFAs are used in the transmission as booster, as pre-amplifier and for LO amplification. And this is only for 5 channels. Please justify quantitatively the benefit in terms of power consumption of such scheme compared with single laser DWDM configurations.

Now it's likely for single-wavelength O-band coherent transceivers to operate without optical amplifications considering the limited fiber loss.

We thank the reviewer for this comment. We agree that the use of 3 amplifiers in the experimental setup is not a good representation of the concept we are proposing. In our concept figure (Fig. 2), the QD-MLL comb source would be integrated with the IQMs. This would remove the need for the first amplifier stage as the power per line is -3 dBm which is sufficient at this stage. It should be noted that every line does not need its own amplifier. Additionally, the concept could make use of more efficient O-band amplifiers such as quantum-dot semiconductor optical amplifier (QD-SOA). We recently demonstrated that a single QD-SOA can be used to operate high speed WDM coherent systems [1]. In this paper, a single SOA was used for 4 lines. In this sense, our scheme does not require 20+ SOAs. Our analysis shows that 7 SOAs can be used to operate the concept presented in Fig 2. The third PDFAs relaxes the launch power requirements thus reducing the four-wave mixing while compensating for the lack of a transimpedance amplifier in our setup.

In order to compare the power consumption of individual laser sources to a comb-based system with SOAs, the following assumptions were made, i) both systems would require the same optical link budget per λ , ii) one SOA was needed for every 4 comb channels, and iii) each individual laser source would consist of a DFB and requires a TEC. The typical values used for the DFBs were a slope efficiency of 0.29 mW/mA, a bias voltage of 1.6 V, and a threshold current of 15 mA [2]. Following the power consumption comparison established in [3], the total link budget for a 10km O-band WDM coherent system with 26 channels would be around 17.4 dBm. This would result in an assumed laser power of 6.2 dBm. Each TEC is assumed to consume around 1.2 W [4].

Operating 26 DFBs with 6.2 dBm each would consume an aggregate of $0.1658 \text{ W} \times 26 \text{ lasers} = 4.31 \text{ W}$. Each laser source would need its own TEC, resulting in 26 TECs total for an additional $26 \times 1.2 \text{ W} = 31.2 \text{ W}$. The total power consumption from the laser sources would be around 35.51 W. On the other hand, a single QD-MLL consumes around 352 mW. A QD-SOA amplifying 4 λ s simultaneously would consume an additional 1.04 W to obtain the same power per channel as the individual lasers. This would result in a total power of $0.352 \text{ W} + 1.04 \text{ W} \times 26/4 = 7.632 \text{ W}$. The number of TECs is also reduced from 26 to 8 as the comb-based system only requires one TEC for all laser channels and 7 for the SOAs. This would result in an estimated TEC power consumption of $7 \times 1.2 \text{ W} = 8.4 \text{ W}$ and a total laser source power of 16.03 W. This represents a 75% reduction in TEC power consumption. This results in a total power consumption reduction of 45.1% when switching to a comb based OFTS as opposed to single-carrier laser sources.

[1] C. St-Arnault, S. Bernal, et. al, "Net 1.6 Tbps (4x400Gbps/ λ) O-Band IM/DD transmission over 2 km using uncooled DFB lasers on the LAN-WDM grid and sub-1V drive TFLN modulators" Optical Fiber Communication Conference (OFC), post-deadline paper, Th4C.6 (2024)

[2] "Aerodiode model 3 (O-band) DFB laser," 2023. Accessed: May 2, 2024. [Online]. Available: <https://www.aerodiode.com/product/1310-nm-laserdiode>

[3] Berikaa, E. et al. Next-Generation O-band Coherent Transmission for 1.6 Tbps 10 km Intra-Datacenter Interconnects. *Journal of Lightwave Technology* (2023).

[4] F. Chang and R. Chen, "Relative cost analysis on IMDD vs coherent for 800G-LR," IEEE P802.3df Task Force Meeting, 2022. Accessed: May 2, 2024. [Online]. Available: https://www.ieee802.org/3/df/public/22_11/chang_3df_01_2211.pdf

2.5. I don't quite get how the adaptive equalizer can remove residual FO up to 20 MHz. Typical adaptive equalizers (e.g., CMA) only focus on amplitude and don't touch on the phase of the symbols. Please specify what exactly the configuration of the 4x4 MIMO equalizer. Does it include also a phase noise compensation process?

We thank the reviewer for the question and apologize for the confusion.

The equalizer used is a time-domain 4x4 real-valued MIMO equalizer (Butterfly form) that uses the LMS algorithm and mean square error between training symbols and the received symbols after the channel as a criterion. The MIMO is further interleaved with a 2nd order phase locked loop that uses the phase error between the training and received symbols and offers fine tracking of the carrier phase and the frequency offset [1]. This equalizer structure has been previously used such as in [2] and enables compensating CD, FO, and phase noise, but with very limited range as seen from our data for FO (0.035 % of the symbol rate). In practical systems, a coarse FO estimator tracks the GHz portion of the FO, while the equalizer does the fine tracking within a few MHz for optimum performance.

[1] M. Torbatian et al., "Performance Oriented DSP for Flexible Long Haul Coherent Transmission," in *Journal of Lightwave Technology*, vol. 40, no. 5, pp. 1256-1272, 1 March1, 2022, doi: 10.1109/JLT.2021.3134155

[2] M. Y. S. Sowailam et al., "770-Gb/s PDM-32QAM Coherent Transmission Using InP Dual Polarization IQ Modulator," in *IEEE Photonics Technology Letters*, vol. 29, no. 5, pp. 442-445, 1 March1, 2017, doi: 10.1109/LPT.2017.2655441.

2.6. The 10% reduction in DSP power consumption with O-band compared with C-band is not very clearly justified. For 10km distance at 56 Gbaud, typical adaptive equalizers with a bit longer than 10-symbol memory length can also compensate CD quite well in the C-band.

We thank the reviewer for this feedback. We modified the main text to better explain this.

Here, we refer to the power consumption of the CD compensation module in the typical C-band coherent pluggable, which uses frequency domain equalizers to compensate for the chromatic dispersion without using an extensively large time-domain filter. For a 56 Gbaud and 10 km transmission, equalizing the CD in the time domain requires 21 taps as calculated using equation 9 in [1]. Although implementing 21 taps is feasible, it still consumes considerable power due to the complexity of the linear convolution. According to [2], the frequency domain CD compensation block consumes 20% of the DSP power consumption and takes up more than 15% of the Die area. Here, the DSP refers to the carrier and data

recovery module (CDR) and signal processing blocks excluding the FEC engines, DACs, and ADCs. Thus, the DSP power itself is a small portion that does not exceed 30% of the entire ASIC power consumption. We assume that the time-domain CD equalizer with 21 taps consumes only 10% of the DSP power consumption for the 10 km case (50% lower than what is current C-band pluggable), and that 10% can be saved if we operate in the O-band dispensing the need for CD compensation.

[1] Seb J. Savory, "Digital filters for coherent optical receivers," *Opt. Express* 16, 804-817 (2008)

[2] C. Fougstedt, O. Gustafsson, C. Bae, E. Börjeson and P. Larsson-Edefors, "ASIC Design Exploration for DSP and FEC of 400-Gbit/s Coherent Data-Center Interconnect Receivers," 2020 Optical Fiber Communications Conference and Exhibition (OFC), San Diego, CA, USA, 2020, pp. 1-3.

2.7. How was the IQ modulator biased? Figure 4c shows central peaks for modulated carriers. It seems like the carriers were not suppressed like conventional QAM modulation.

We thank the reviewer for the question, and we apologize for the confusion. The modulator was biased manually by tuning the individual phase shifters on the I and Q branches of the IQM. The bias was optimized by measuring the resulting BER and constellation during the experiment. This method was used due to a lack of an automatic bias controller and could lead to some bias errors. Additionally, the extinction ratio of the modulator was relatively low, leading to some additional leaking of the carrier. We added this explanation to the methods section.

2.8. For the IMDD transmission, how much was the bandwidth of the TBF?

We thank the reviewer for this question. In the IMDD experiment, two TBF were used. The bandwidth of the TBF after the comb source was set to around 180 GHz. This allowed 3 comb lines to be bulk-modulated at once. This prevented saturating the PDFAs while still allowing for a neighbor on each side of the channel under test. The bandwidth of the TBF before the receiver was set to around 58 GHz. This allowed the second TBF to act as a perfectly centered demux that mitigated the effects of neighboring channels at the receiver. We added a clarification to the methods section to address this.

2.9. Again for IMDD, was the 2dBm ROP per channel or per 3 channels? And how much was it for coherent ROP? In fact, considering PDFAs are used for the signal, one should really show OSNR than ROP values as the signals are additive noise limited rather than receiver noise limited.

We thank the reviewer for the question and comment. The 2dBm ROP was for a single channel under test. For IMDD, a second TBF was required which reduced the link budget by an additional 6 dBm. For the coherent case, the ROP was kept constant around 5 dBm. This

consisted mostly of the LO power. We modified the Methods section to clarify this.

Other comments:

2.10. In the Abstract, when claiming 'highest capacity C-band systems reported to date', please add condition that it is compared with QD-MLL based C-band systems. Record transmission rate of C-band systems in general is much higher.

We thank the reviewer for the correction. We addressed the statement and added different comb types as suggested by Reviewer 1 to the introduction.

2.11. Some typos are spotted in the introduction, 'Gpbs'  'Gbps'.

We thank the reviewer for pointing out the error. We fixed them in the main text.

2.12. The linewidth color bar shown in Fig. 3a may be better shown with log scale, as it is more interesting to focus on low-linewidth region below 500 kHz, which is all blue in the current figure.

We thank the reviewer for this suggestion. We modified the figure accordingly as seen below.

Figure 1: **QD-MLLs characterization.** Different comb states were characterized and compared through their **a** linewidth, ...

2.13. In first paragraph of Sec. 4.1, some figure numbers are missing (Fig. b, Fig. c).

We thank the reviewer for pointing out this mistake. It has been fixed in the main text.

2.14. On page 21, The sentence 'By increasing the ROP of the DFB from 2 dBm to 2.7 dBm, ...' is confusing.

We thank the reviewer for pointing out this confusion. We modified the main text to clarify that the received optical power here was for the photodetector. The ROP was increased by changing the output power of the DFB laser.

2.15. On page 22, there is a statement of "The signal bandwidth cannot be larger than the spacing between comb lines due to the penalty of inter symbol interference (ISI)". I am not sure if ISI is the correct term here.

We thank the reviewer for pointing this out. We agree that ISI is not the correct term here. The limiting factor here is the linear cross talk between WDM channels. We updated the text to correct this mistake.

2.16. In the first paragraph of Sec. 5, 100 bps/lambda should be 100 Gbps/lambda.

We thank the reviewer for pointing out this mistake. It has been corrected in the main text.

Reviewer 3:

The submitted manuscript discusses the use of frequency combs for ultra-high speed intra-datacenter communications.

The work is generally very relevant wrt next generation intra DCN.

The manuscript is very well written and concise, however, there are a number of aspect the authors could improve.

We thank the reviewer for the summary of the work and the kind words.

3.1- it would be good to mention the fiber parameters. Especially relevant are exact position of dispersion zero, dispersion slope and PMD value.

We thank the reviewer for this suggestion. We added these parameters to the methods sections where applicable.

For the simulations the values are as follows:

1. The zero-dispersion wavelength was positioned at the center of the channel plan, corresponding to 1299 nm. This is the worst scenario to induce fiber nonlinear behavior.
2. The dispersion slope used corresponds to an ITU-T G.652 fiber, which in this case was SMF-28. The value was $0.09 \text{ ps/nm}^2\text{-km}$.
3. The PMF coefficient used was 0.05 ps/sqrt(km) which is commonly found in fibers.

3.2- in section 2.1 IM/DD it would be good to already mention FWM as an important impairment hindering tighter channel grids in IM/DD high speed transmission in the O-Band.

Also some discussions could be added on recently proposed ways of improving FWM penalties (e.g. orthogonal polarization launch of neighboring channels)

We thank the reviewer for the comment and suggestion. We added a statement introducing FWM in section 2.1 as an issue for tight IMDD channel grids. We also added the following discussion to section 4.2 to indicate ways to improve the IMDD NL effects. Lastly, we added a section in the Supplemental to further discuss orthogonal polarization launch (see reviewer response 3.6).

4.2:

To reduce FWM in IMDD DWDM systems, multiple solutions have been proposed such as using unequal channel spacings [1] or using orthogonal polarization launch of neighboring channels [2]. FWM penalties will be impacted by PMD because this leads to depolarization along the transmission line. As PMD increases, the FWM penalty is reduced and forms a distribution due to the randomness of total differential group delay (DGD) and the orientation of the principal state of the polarization vector relative to the launch polarization [3]. Note that we indeed included PMD in our simulations. When only four channels are launched the use of a YXXY polarization at the input decreases the FWM impairment. This has been proposed to implement the 800G-LR4 Ethernet node. This can be generalized to 8 channels as XYYXXYYX or to 26 channels as shown in supplementary section 7.

[1] I. Rasheed, M. Abdullah, S. Mehmood and M. Chaudhary, "Analyzing the non-linear effects at various power levels and channel counts on the performance of DWDM based optical fiber communication system," 2012 International Conference on Emerging Technologies, Islamabad, Pakistan, 2012, pp. 1-5, doi: 10.1109/ICET.2012.6375446.

[2] Liu, Xiang & Fan, Qirui. (2023). Inter-Channel FWM Mitigation Techniques for 800G-LR4, 1.6T-LR8, 400G-ER4 and 5G Fronthaul Applications Based on O-band WDM. Journal of Lightwave Technology. PP. 1-10. 10.1109/JLT.2023.3316008.

[3] J. P. Gordon and H. Kogelnik, "PMD fundamentals: polarization mode dispersion in optical fibers," Proc Natl Acad Sci U S A, vol. 97, no. 9, pp. 4541-50, Apr 25 2000, doi: 10.1073/pnas.97.9.4541.

3.3- it would be interesting to add a short discussion about system parameter differences between single channel lasers and QD-MLLs such as total power consumption and heat dissipation.

We thank the reviewer for the suggestion. We expanded the power consumption comparison between single lasers and the comb as requested in reviewer comment 2.4. Our calculations revealed a 45% decrease in laser and TEC power consumption when implementing a comb-based system. This analysis was added to section 5 of the paper. However, we believe that a QD-MLL heat dissipation discussion is outside the scope of this paper as the focus is on the transmission system performance of using comb-based laser sources.

3.4- in section 3 the power conversion efficiency is mentioned to be high. It would be good to quantify (and compare to "normal" lasers)

We thank the reviewer for the suggestion. We added a reference and summary to our previous work [1] which measures the efficiency of the QD-MLL comb to section 3. Additionally, we expanded the power consumption comparison between single lasers and the QD-MLL comb as requested in reviewer comment 2.4 to section 5.

[1] Dumont, M.; Liu, S.; Kennedy, M.J.; Bowers, J. High-Efficiency Quantum Dot Lasers as Comb Sources for DWDM Applications. *Appl. Sci.* 2022, 12, 1836. <https://doi.org/10.3390/app12041836>

3.5- in Fig. 4 some constellations show non-round (i.e. nonlinear) shape. Can you give an explanation?

We thank the reviewer for this question. The clusters of the constellation do not appear perfectly rounded due to the equalizer-enhanced phase noise. This is because the noise becomes slightly correlated between the I and Q quadratures as both go through the MIMO equalizer. Since the equalizer used is a 4x4 MIMO with real-valued coefficients, the I and Q have cross terms which might lead to noise correlation, but it can also fix the IQ skew. Additionally, the slight curve on some of the constellations comes from a small imbalance in the Q and I arms. This imbalance is likely caused by a probing issue encountered during the experimental process resulting in slightly different RF signal amplitudes being applied to the IQM arms.

3.6- in Fig. 6 a comparison of NL penalty is shown. In recent research of 1.6TbE+ systems, orthogonal (polarization) launch is discussed to significantly reduce FWM penalties. Could you please add a discussion and also explain whether this could also be realized in your comb system setup. Also the effect of PMD will have a significant impact on FWM penalties as it leads to depolarization along the transmission line. This may be difficult to evaluate experimentally, though. Did you see any temporal variations of the FWM penalty?

We thank the reviewer for the question and suggestion. We added a small discussion in section 4.2 as shown in reviewer response 3.2 to discuss orthogonal polarisation launch. We also added the following section to the supplemental:

Supp:

Using orthogonal polarization launch of neighboring channels has been proven to reduce NL effects in DWDM systems [1]. Figure S12 shows the simulated improvement when using a XYXXYYXXYYXXYYXXYYXXYYXX polarization configuration. Our results show a 3 dB improvement when using this method in our comb system with 26 lines over 10 km with a total optical power input into the fiber of 6 dBm. This method could be applied to an

integrated solution by using on-chip polarisation rotators [2-3] to reduce the effects of FWM in DWDM IMDD systems. Alternatively, this solution can also be applied to a pluggable scheme (see Supplemental Section 3) by using external polarisation rotators as well as polarisation maintaining fibers before and after the mux and demux, respectively. While this penalty reduction is significant, it is not sufficient to surpass the coherent tolerance to FWM in comb-based OFTS. Additionally, our results show that PMD does not significantly impact the system performance in this use case.

Figure S12: Calculated BER penalty as a function of total input power into the fiber for PAM4 over 10km. A comb with 25 lines was used as transmitter with a line spacing of 58 GHz.

[1] Liu, Xiang & Fan, Qirui. (2023). Inter-Channel FWM Mitigation Techniques for 800G-LR4, 1.6T-LR8, 400G-ER4 and 5G Fronthaul Applications Based on O-band WDM. *Journal of Lightwave Technology*. PP. 1-10. 10.1109/JLT.2023.3316008.

[2] Sun, B., Morozko, F., Salter, P.S. et al. On-chip beam rotators, adiabatic mode converters, and waveplates through low-loss waveguides with variable cross-sections. *Light Sci Appl* 11, 214 (2022). <https://doi.org/10.1038/s41377-022-00907-4>

[3] H. Guan, Q. Fang, G. -Q. Lo and K. Bergman, "High-Efficiency Biwavelength Polarization Splitter-Rotator on the SOI Platform," in *IEEE Photonics Technology Letters*, vol. 27, no. 5, pp. 518-521, 1 March 1, 2015, doi: 10.1109/LPT.2014.2384451.

3.7- it would be interesting to mention also which exact DSP has been used. Please give some details about filter lengths, etc. In Fig. 9 no equalizer (FFE/DFE) is shown. Is that correct? Or is it included in the MIMO eq? Later on it is referred to "a conventional DSP" . Please specify in more details.

We thank the reviewer for the comment. As also requested in reviewer comment 2.5, we added more details on the exact DSP stacks used in the main text.

The equalizer used is a time-domain 4x4 real-valued MIMO equalizer (Butterfly form) that uses the LMS algorithm and mean square error between training symbols and the received symbols after the channel as a criterion. The MIMO is further interleaved with a 2nd order phase locked loop that uses the phase error between the training and received symbols and offers fine tracking of the carrier phase and the frequency offset [1]. This equalizer structure has been previously used such as in [2] and enables compensating CD, FO, and phase noise, but with very limited range as seen from our data for FO (0.035 % of the symbol rate). In practical systems, a coarse FO estimator tracks the GHz portion of the FO, while the equalizer does the fine tracking within a few MHz for optimum performance.

[1] M. Torbatian et al., "Performance Oriented DSP for Flexible Long Haul Coherent Transmission," in *Journal of Lightwave Technology*, vol. 40, no. 5, pp. 1256-1272, 1 March 2022, doi: 10.1109/JLT.2021.3134155

[2] M. Y. S. Sowailam et al., "770-Gb/s PDM-32QAM Coherent Transmission Using InP Dual Polarization IQ Modulator," in *IEEE Photonics Technology Letters*, vol. 29, no. 5, pp. 442-445, 1 March 2017, doi: 10.1109/LPT.2017.2655441.

3.8- In Fig. 10 an FFE is shown. In the text offline DSP is mentioned. Please give details.

We thank the reviewer for this feedback. We added more details on the IMDD DSP used in the methods section to explain the steps taken to perform offline DSP as is accepted in literature [1-3].

The PAM symbols are generated from a random binary sequence. The signal is filtered by a raised cosine (RC) pulse shaping filter to limit its bandwidth to 2 samples per symbol (sps) and then resampled to the DAC sampling rate. We used a single pre-emphasis filter to pre-compensate for the frequency response of the DAC. At the receiver, the capture from the photodetector is digitized by an ADC such that the DSP can be performed offline.

[1] E. Berikaa, M. S. Alam and D. V. Plant, "Beyond 300 Gbps Short-Reach Links Using TFLN MZMs With 500 mVpp and Linear Equalization," in *IEEE Photonics Technology Letters*, vol. 35, no. 3, pp. 140-143, 1 Feb. 2023, doi: 10.1109/LPT.2022.3227085

[2] W. Li et al., "Thin-Film BTO-Based MZMs for Next-Generation IMDD Transceivers Beyond 200 Gbps/ λ ," in *Journal of Lightwave Technology*, vol. 42, no. 3, pp. 1143-1150, 1 Feb. 2024

[3] E. Berikaa et al., "TFLN MZMs and Next-Gen DACs: Enabling Beyond 400 Gbps IMDD O-Band and C-Band Transmission," in *IEEE Photonics Technology Letters*, vol. 35, no. 15, pp. 850-853, 1 Aug. 2023, doi: 10.1109/LPT.2023.3285881.

3.9- In the supplement a working temperature of 42°C is mentioned. Apart from wavelength drift, what effect has a change in temperature on optical power?

We thank the reviewer for this question. We noticed a slight increase in the power per line of around 0.5 dB for the edge channels when increasing the laser temperature from 40°C to

45°C. The power of the other channels was not really affected. Other laser parameter changes over temperature such as linewidth, or RIN were not investigated for this study as we believe that laser performance as a function of temperature is outside the scope of the paper given the focus on transmission system performance. Additionally, a TEC was always used in the assumptions and calculations.

3.10- Could you add an axis for temperature variation in S7 (e.g. second x-axis)?

We thank the reviewer for this suggestion to make our figure clearer. We added the temperature axis to the requested figure as seen below.

Fig S2: **Calculated SNR penalty as a function of a temperature-induced emission frequency shift.** PAM4 and DP-16QAM modulation formats are shown at 56 GBaud after 10km in the O-band. A comb with 5 lines is used as a transmitter. A line spacing of 58 GHz is used in all cases.

REVIEWERS' COMMENTS

Reviewer #1 (Remarks to the Author):

I am happy overall with the responses and the changes - to the title and throughout the paper as well as more qualified arguments/claims. I would be happy to see this paper published now, and would only request 2 more small changes (I do not need to re-review the MS - it can be done by the Editor:

- addition of the new OFC'2024 references, to increase the timeliness, on O-band transmission inc papers from KDDI who are currently leading the world on this...esp the PD paper by D Elson et al,

"Continuous 16.4-THz Bandwidth Coherent DWDM Transmission in O-band using a Single Fibre Amplifier System",

as well as any other relevant references

- explanation of why the measurement of phase noise is so difficult - in rebuttal under 1.4 - there should be a paragraph added to the paper explaining why the measurement of phase noise is such a challenge and why it has not been addressed, and perhaps a reference to others who have done it...eg work at UCL by Deakin, Sohanpal, Zhixin Liu et al? or any other work that's relevant!

Reviewer #2 (Remarks to the Author):

I have gone through the authors' response and the revised manuscript. I would like to acknowledge the authors' efforts in addressing my comments in a careful and proper way. The explanations and the corresponding revisions are reasonable and convincing. Particularly, for the sliding window and bulk modulation comment, the additional numerical analyses provide clear and strong arguments for the adopted scheme. I don't have further comments and I am happy to recommend accepting the paper for publication.

Reviewer #3 (Remarks to the Author):

The authors have picked up all my remarks from the first round of revisions and adequately revised their manuscript.

Responses to Reviewer comments #2

Reviewer 1:

Reviewer #1 (Remarks to the Author):

I am happy overall with the responses and the changes - to the title and throughout the paper as well as more qualified arguments/claims. I would be happy to see this paper published now, and would only request 2 more small changes (I do not need to re-review the MS - it can be done by the Editor:

We thank the reviewer for taking the time to review the manuscript again.

- addition of the new OFC'2024 references, to increase the timeliness, on O-band transmission inc papers from KDDI who are currently leading the world on this...esp the PD paper by D Elson et al,

"Continuous 16.4-THz Bandwidth Coherent DWDM Transmission in O-band using a Single Fibre Amplifier System",

as well as any other relevant references

We thank the reviewer for this suggestion. We have added the following citations [1-3] to the manuscript in order to increase its timeliness as suggested. These consists of 2024 OFC post deadline papers which show the latest demonstrations of O-band WDM optical fiber transmission systems.

[1] D. J. Elson, et al. "Continuous 16.4-THz Bandwidth Coherent DWDM Transmission in O-band using a Single Fibre Amplifier System," in Optical Fiber Communication Conference (OFC) 2024, Technical Digest Series (Optica Publishing Group, 2024), paper Th4A.2.

[2] C. St-Arnault, et al. "Net 1.6 Tbps ($4 \times 400\text{Gbps}/\lambda$) O-Band IM/DD Transmission Over 2 km Using Uncooled DFB Lasers on the LAN-WDM grid and Sub-1V Drive TFLN Modulators," in Optical Fiber Communication Conference (OFC) 2024, Technical Digest Series (Optica Publishing Group, 2024), paper Th4C.6.

[3] S. Misak, et al., "400 Gbps/ λ DP-16QAM O-band Link with SiP TX and RX PICs using only Heterogeneously Integrated Lasers and SOAs for Optical Gain," in Optical Fiber Communication Conference (OFC) 2024, Technical Digest Series (Optica Publishing Group, 2024), paper Th4C.5.

- explanation of why the measurement of phase noise is so difficult - in rebuttal under 1.4 - there should be a paragraph added to the paper explaining why the measurement of phase noise is such a challenge and why it has not been addressed, and perhaps a reference to others who have done it...eg work at UCL by Deakin, Sohanpal, Zhixin Liu et al? or any other work that's relevant!

We thank the reviewer for this suggestion and apologise for the confusion in our rebuttal. In response 1.4, we would like to clarify that the phase noise of the QD-MLL used in this experiment had already been studied and published [1] previously. We therefore reference this work in the manuscript without presenting the results directly again. Additionally, we decided not to include a phase noise comparison between our QD-MLL and other comb-based laser sources in order to keep the focus of the paper on a specific type of comb laser source. This focus agrees well with our results as well as the reviewer suggestions. We believe an in-depth comparison of the phase noise from different comb laser sources could be the focus of a follow-up paper.

[1] Zibar, Darko; Razumov, Aleksandr; Heebøll, Holger; Dumont, Mario; Terra, Osama; Dong, Bozhang; et al. (2023). Subspace tracking for phase noise source separation in frequency combs. *Optica Open*. Preprint. <https://doi.org/10.1364/opticaopen.24155511.v1>

Reviewer 2:

Reviewer #2 (Remarks to the Author):

I have gone through the authors' response and the revised manuscript. I would like to acknowledge the authors' efforts in addressing my comments in a careful and proper way. The explanations and the corresponding revisions are reasonable and convincing. Particularly, for the sliding window and bulk modulation comment, the additional numerical analyses provide clear and strong arguments for the adopted scheme. I don't have further comments and I am happy to recommend accepting the paper for publication.

We thank the reviewer for taking the time to review the manuscript again and for the kind words.

Reviewer 3:

Reviewer #3 (Remarks to the Author):

The authors have picked up all my remarks from the first round of revisions and adequately revised their manuscript.

We thank the reviewer for taking the time to review the manuscript again.